

# Intercomparison of AIRS and HIRDLS stratospheric gravity wave observations

Catrin I. Meyer[1], Manfred Ern[2], Lars Hoffmann[1], Quang Thai Trinh[2], and M. Joan Alexander[3]

[1]Jülich Supercomputing Centre (JSC), Forschungszentrum Jülich, Jülich, Germany
[2]Institut für Energie- und Klimaforschung (IEK-7), Forschungszentrum Jülich, Jülich, Germany
[3]NorthWest Research Associates, Inc., CoRA Office, Boulder, CO, USA

*Correspondence to:* C. I. Meyer (cat.meyer@fz-juelich.de)

**Abstract.** We investigate stratospheric gravity wave observations by the Atmospheric InfraRed Sounder (AIRS) aboard NASA's Aqua satellite and the High Resolution Dynamics Limb Sounder (HIRDLS) aboard NASA's Aura satellite. AIRS operational temperature retrievals are typically not used for studies of gravity waves, because their horizontal resolution is rather limited. This study uses data of a high-resolution retrieval which provides stratospheric temperature profiles for each individual satellite

footprint. Therefore the horizontal sampling of the high-resolution retrieval is nine times better than that of the operational retrieval. HIRDLS provides 2D spectral information of observed gravity waves in terms of along-track and vertical wavelengths. AIRS as a nadir sounder is more sensitive to short horizontal wavelength gravity waves and HIRDLS as a limb sounder is more sensitive to short vertical wavelength gravity waves. Therefore HIRDLS is ideally suited to complement AIRS observations. A calculated momentum flux factor indicates that the waves seen by AIRS contribute significantly to momentum flux, even if

the AIRS temperature variance may be small compared to HIRDLS. The stratospheric wave structures observed by AIRS and HIRDLS agree often very well. Case studies of a mountain wave event and a non-orographic wave event demonstrate that the observed phase structures of AIRS and HIRDLS are conform. AIRS has a coarser vertical resolution, which results in an attenuation of the amplitude and coarser vertical wavelengths compared to HIRDLS. However, AIRS has a much higher horizontal resolution and the propagation direction of the waves can be clearly identified in geographical maps. The horizontal orientation

of the phase fronts can be deduced from AIRS 3D temperature fields. This is a restricting factor for gravity wave analyses of limb measurements. Additionally, temperature variances with respect to stratospheric gravity wave activity are compared on a statistical basis. The complete HIRDLS measurement period from January 2005 to March 2008 is covered. The seasonal and latitudinal distributions of gravity wave activity as observed by AIRS and HIRDLS fit well. A strong annual cycle at mid and high latitudes is found in time series of gravity wave variances at 42 km, which has during wintertime its maxima and during

summertime its minima. During austral wintertime at 60°S the variability is largest. Variations in the zonal winds at 2.5 hPa are associated with large variability in gravity wave variances. Altogether, gravity wave variances of AIRS and HIRDLS are conform and complementary to each other. Thereby large parts of the gravity wave spectrum are covered by joint observations. This opens up fascinating vistas for future gravity wave research.



# 1 Introduction

By driving the general circulation, the thermal structure and middle atmosphere chemistry are influenced significantly by atmospheric gravity waves (Lindzen, 1973; Holton, 1982, 1983; McLandress, 1998; Fritts and Alexander, 2003; Eyring et al., 2007). The generation and propagation of gravity waves depends on the sources and atmospheric conditions. Gravity waves are

primarily generated due to orography, like mountain waves (Smith, 1985; Durran and Klemp, 1987; Nastrom and Fritts, 1992; Dörnbrack et al., 1999), and as a result of deep convection (Pfister et al., 1986; Tsuda et al., 1994; Alexander and Pfister, 1995; Vincent and Alexander, 2000). Additionally, gravity waves originate due to body forcing, which comes along with localized wave dissipation, and wave-wave interaction (Fritts and Alexander, 2003; Vadas et al., 2003) and due to wind shear, adjustment of unbalanced flows near jet streams and frontal systems (Fritts and Nastrom, 1992; Wu and Zhang, 2004; Plougonven et al.,

2003). Most global atmospheric models use gravity wave parameterizations because gravity waves are small-scale phenomena and cannot be resolved or are only poorly resolved in the models. Satellite observations are well suited to validate gravity wave parametrization schemes of general circulation models. In addition, characteristics of gravity waves can be investigated in global studies with satellite observations (Geller et al., 2013).

Fetzer and Gille (1994) were the first to demonstrate that satellite remote sensors can observe gravity waves. The number of

instruments with sufficient spatial resolution to observe gravity waves has increased over the last years. An important limitation of satellite observations is that each instrument type can only detect a certain part of the full vertical and horizontal wave number spectrum of gravity waves. Wu et al. (2006), Preusse et al. (2008), and Alexander et al. (2010) give an overview and comparison of different observation methods and the range of detectable vertical and horizontal wavelengths. Advantages and disadvantages of limb measurements vary in contrast to nadir instruments. Limb instruments have a good vertical resolution,

which leads to high sensitivity to short vertical wavelength waves. However, the sensitivity for short horizontal wavelengths is reduced due to the limited horizontal resolution of current limb sounders (Preusse et al., 2009b). Furthermore, a single measurement track can not be used to identify the horizontal propagation direction of the waves. Nadir instruments observe only gravity waves with long vertical wavelengths, but the horizontal resolution is better in contrast to limb instruments.

For studies of atmospheric gravity waves AIRS radiance measurements are appropriate. The long-term time series of AIRS

radiance measurements offers the opportunity to study gravity wave occurrence frequencies and other characteristics climatologically and on a global scale (Gong et al., 2012; Hoffmann et al., 2013, 2014). AIRS operational temperature retrievals are typically not used for gravity wave research. A main drawback is their limited horizontal resolution related to the cloud-clearing procedure. This procedure facilitates retrievals in the troposphere by combining radiance measurements of $3 \times 3$ footprints to reconstruct a single cloud-free spectrum. This causes a substantial loss of horizontal resolution. Nevertheless, stratospheric 3D

temperature fields with a high spatial resolution can be retrieved from AIRS radiances. The AIRS high-resolution retrieval of Hoffmann and Alexander (2009) provides a temperature data set which is considered optimal for stratospheric gravity wave studies. Meyer and Hoffmann (2014) performed a comparison between the AIRS high-resolution stratospheric temperature retrieval, the AIRS operational Level-2 data, and the ERA-Interim reanalysis (Dee et al., 2011) on the basis of nine measurement years (2003–2011). That study showed that the AIRS high-resolution retrievals reproduce mean and standard deviations



of ERA-Interim stratospheric temperatures with good accuracy. Zonal averages incline to be mostly below $\pm\,2\,\mathrm{K}$. Sato et al. (2016) used the AIRS high-resolution retrievals to study interactions of gravity waves with the El Niño-Southern Oscillation (ENSO). Tsuchiya et al. (2016) investigated interactions of gravity waves with the Madden-Julian Oscillation (MJO) using the same data set. Ern et al. (2017) and Wright et al. (2017) applied 3D spectral analysis techniques to the AIRS high-resolution retrievals and estimate thereby directional gravity wave momentum flux.

By using the limb sounding technique, HIRDLS is sensitive to short vertical wavelength gravity waves and is therefore ideally suited to complement AIRS observations. HIRDLS temperature observations have been widely used to study the global distribution of gravity waves. In particular, absolute gravity wave momentum fluxes are derived from information about gravity wave vertical and horizontal wavelengths (Alexander et al., 2008; Wright et al., 2010; Ern et al., 2011). Based on these momentum fluxes, the intermittency in gravity wave global distributions was studied (e.g., Hertzog et al., 2012; Wright et al., 2013), as well as the interaction of gravity waves with the background circulation (e.g., Ern et al., 2014, 2015). In addition Geller et al. (2013) used HIRDLS data to compare gravity wave momentum fluxes in models and those derived from observations. The main advantage of HIRDLS is that 2D spectral information of observed gravity waves is provided in terms of along-track and vertical wavelengths. This information has been utilized for studying the average spectrum of gravity waves in different regions (e.g., Lehmann et al., 2012; Ern and Preusse, 2012; Trinh et al., 2016). We will use this information here to comprehensively compare AIRS and HIRDLS gravity wave observations, which is the main aim of our study.

The AIRS and HIRDLS instrument characteristics and the gravity wave observations are introduced in Sect. 2. We explain the detrending method and noise corrections that we used to estimate gravity wave variances from AIRS and HIRDLS observations. Further, nadir and limb observation geometries are compared regarding their sensitivities to gravity horizontal and vertical wavelengths. In Sect. 3 we present case studies of coincident AIRS and HIRDLS gravity wave observations and comparisons of time series of gravity wave variances from AIRS and HIRDLS during 2005 to 2008. In addition, the influence of the AIRS observational filter is investigated. In Sect. 4 we will draw conclusions and give an outlook.

## 2  Data and methods

### 2.1  AIRS and HIRDLS observations and temperature retrievals

The Aqua satellite is part of NASA's Earth Observing System and the first satellite in the A-Train constellation. The flight altitude of Aqua is $705\,\mathrm{km}$ and it performs in a sun-synchronous polar orbit with an inclination of $98°$ and a period of $99\,\mathrm{min}$. On-board NASA's Aqua satellite six instruments are included and one of them is the Atmospheric InfraRed Sounder (AIRS) (Aumann et al., 2003; Chahine et al., 2006). Thermal emissions of atmospheric properties in the nadir and sub-limb geometry are measured by AIRS. 14.5 orbits are completed by AIRS per day. At 1:30 am (descending orbit) and 1:30 pm (ascending orbit) local time the equator crossing occurs. AIRS has across-track scanning capabilities. One scan captures $1780\,\mathrm{km}$ ground distance with 90 individual footprints. The scans are performed in $2.667\,\mathrm{sec}$ and the along-track distance is $18\,\mathrm{km}$. Granules of six minutes measurement time, i.e., 135 scans or 12150 footprints, are accumulated in the AIRS measurements. 2.9 million radiance spectra are globally detected by AIRS within one day. The measurement coverage of the AIRS instrument is almost





complete since the observations started in September 2002. The analysis of this study is based on measurements during January 2005 to March 2008, which is the measurement period of HIRDLS.

Aqua carries different instruments, which measure radiation in the near and mid infrared and the microwave spectral re-

gions (Aumann et al., 2003; Gautier et al., 2003; Lambrigtsen, 2003). Several retrieval algorithms transform the calibrated radiances into geophysical quantities (Susskind et al., 2003; Goldberg et al., 2003). The original resolution of the AIRS radiance measurements (Level-1 data) is reduced during the operational retrieval (Level-2 data) by a factor of $3\times3$ (along-track $\times$ across-track). Thereby the retrievals are extended into the troposphere and cloud clearing is performed (Barnet et al., 2003; Susskind et al., 2003; Cho and Staelin, 2006). Several linear and nonlinear operations on the infrared and microwave channels

are required for the cloud clearing algorithm. The algorithm performs on blocks of $3\times3$ AIRS footprints. The clearest field of view in the $3\times3$ block is selected, and a single cloud-cleared infrared spectrum for the block is computed (Cho and Staelin, 2006). Validation of AIRS operational retrievals for the troposphere provide an accuracy which is nearby the anticipated absolute accuracy of 1 K root mean square over a 1 km layer (Fetzer et al., 2003; Divakarla et al., 2006; Tobin et al., 2006). A root mean square deviation of 1.2 and 1.7 K is found in the troposphere and lower stratosphere, respectively, by comparing AIRS

with radiosondes (Divakarla et al., 2006).

A high-resolution retrieval scheme for stratospheric temperatures based on AIRS radiance measurements was developed by Hoffmann and Alexander (2009). This retrieval scheme provides a temperature profile for each individual footprint, corresponding in a horizontal sampling which is $3 \times 3$ times better than the operational retrieval data provided by NASA. While the operational retrievals are tightly constrained in the stratosphere, the high-resolution retrieval configuration offers an optimal

opportunity for gravity wave analyses, because spatial resolution and retrieval noise are balanced in the results by an optimized retrieval configuration. The altitude range of the retrieval is from 10 to 70 km with a 3 km sampling below 60 km altitude and 5 km above. In the stratosphere the high-resolution retrieval has a vertical sampling which is like the AIRS operational retrieval grid. Based on the assumption of hydrostatic equilibrium and using a given reference pressure, the pressure calculated, whereas the temperature is retrieved. In the altitude range between 20 and 60 km the noise of the high-resolution retrieval is about 1.4

to 2.1 K and the total retrieval error, which includes several systematic errors, is 1.6 to 3.0 K. In this altitude range the retrieval achieves the most reliable results, which is indicated by the retrieval diagnostics. There are about 5–6 degrees of freedom for signal in the retrieved profiles.

The retrieval setup of the AIRS high-resolution retrieval distinguishes between day- and nighttime conditions. The Juelich Rapid Spectral Simulation Code (JURASSIC) model (Hoffmann and Alexander, 2009) is used for radiative transfer calcu-

lations. This model assumes local thermodynamic equilibrium (LTE), which restricts the study of daytime measurements to the 15 μm channels. The 4.3 μm channels are affected by non-LTE effects due to solar excitation of $CO_2$ molecules (de Souza-Machado et al., 2007; Strow et al., 2006). Non-LTE effects are not noticed in nighttime measurements of AIRS. Therefore the retrieval uses both wavebands. Lower retrieval noise and better vertical resolution of the nighttime retrievals compared to the daytime retrievals is the consequence. The data in this study was split in day- and nighttime depending on

the solar zenith angle. The retrievals consider values larger than 108° as nighttime data. Note that especially throughout polar summer at high latitudes this limitation leads to data gaps.



The High Resolution Dynamics Limb Sounder (HIRDLS) is a 21 channel infrared limb scanning radiometer aboard NASA's Aura satellite (Gille et al., 2003, 2008). The A-Train constellation of NASA satellites includes Aura, too. Therefore AIRS and HIRDLS cross the same geographic locations within a few minutes. Aura was launched on 15 July 2004 in a sun-synchronous polar orbit. Aura has an inclination of 98° at a flight altitude of 705 km. The line of sight of HIRDLS is fixed to an azimuth of -47° concerning the orbit plane. Therefore a latitudinal coverage of about 63°S to 80°N occurs. Along-track distances between subsequent altitude profiles are down to only 100 km because the line of sight of HIRDLS is fixed. This remarkably fine along-track sampling offers a great opportunity for the analysis of gravity waves. Multiple thin spectral channels of the 15 μm $CO_2$ infrared emissions are used to retrieve atmospheric temperatures. The fractional cover-up of HIRDLS field of view induces perturbations of the measured atmospheric limb radiances, which have been eliminated (Gille et al., 2008). Temperature retrievals are provided for January 2005 to March 2008. HIRDLS measures in an altitude range between the tropopause region and the upper mesosphere on a pressure grid with 121 levels. Between 13 and 60 km the vertical field of view of the instrument is 1 km which is achieved as vertical resolution from the measured temperature altitude profiles (Gille et al., 2008). Our analysis uses retrieval products obtained with processing software version 6. HIRDLS temperature retrievals are carefully validated. Comparisons between HIRDLS and SABER and HIRDLS and ECMWF temperatures indicate that HIRDLS has a warm bias at the tropical tropopause. In the stratosphere HIRDLS temperatures are within 1 K of ECMWF temperatures, within 1–2 K of Microwave Limb Sounder temperatures, and within 2 K of lidar temperatures (Gille et al., 2011).

## 2.2 Removal of background signals to extract gravity wave information

This paper partly focuses on statistical comparisons of temperature variances related to stratospheric gravity wave activity. The total variance ($\sigma_{tot}^2$) of the satellite temperature measurements is typically consisting of three components: the variance of gravity waves ($\sigma_{gw}^2$), of background signals ($\sigma_{bg}^2$), and of noise ($\sigma_{noise}^2$).

$$\sigma_{tot}^2 = \sigma_{gw}^2 + \sigma_{bg}^2 + \sigma_{noise}^2 \qquad (1)$$

To eliminate the background signals from the temperature measurements and to receive gravity wave signals a detrending procedure is necessary. Latitudinal large-scale temperature gradients and planetary wave activity are linked with the background signals. The removal of background signals in AIRS temperature measurements follows the detrending method described by Wu (2004), Eckermann et al. (2006), and Alexander and Teitelbaum (2007). A fourth-order polynomial fit in the across-track direction is used in this method for defining the background. Perturbations are calculated by subtracting the polynomial fit from the raw brightness temperature data. Here we transferred the method to temperature retrievals and applied the fit independently for each altitude. Note that this procedure suppresses strongly wave fronts which are parallel to the across-track direction and which cover large fractions of each scan. This effect can possibly be reduced if the background is smoothed along-track. In the case of extreme latitudinal gradients in the temperature fields, e.g., at the polar vortex edge, problems can be introduced by smoothing. Therefore along-track smoothing was not considered here.

The background removal applied to HIRDLS temperatures comprises several steps. For a fixed latitude and altitude, the data set is subdivided into overlapping time windows of 31 days length. For these 31-day time windows, the zonal mean temperature



and trend are removed, and 2D spectra in longitude and time are estimated. By back-transformation of these spectra for the spectral components exceeding an amplitude threshold, the contribution of planetary waves with zonal wavenumbers up to

6 and periods as short as about 1.4 days is calculated for the precise location and time of each HIRDLS observation, and subtracted. Further, the altitude profiles are vertically filtered in order to remove oscillations with vertical wavelengths longer than about 25 km. The whole procedure is described in more detail in Ern et al. (2011). At the end of the procedure quasi-stationary zonal wavenumbers 0–4 are subtracted to remove the significant tidal modes. Thereby ascending and descending orbits are distinguished (Ern et al., 2013). The final altitude profiles of temperature fluctuations thus obtained are traced back

to mesoscale gravity waves.

It is difficult and always some kind of trade-off to distinguish in observations between planetary and gravity waves. Therefore for both AIRS and HIRDLS a minor contribution of the background variances is caused by gravity waves, depending on the method of background removal. For AIRS, the background may contain minor contributions of gravity waves with long horizontal wavelength, while for HIRDLS the background will contain minor contributions due to gravity waves with long

vertical wavelengths. Still, at most latitudes the background variances will be dominated by global-scale waves. The variances are calculated from the fluctuations relative to a zonal average for a fixed altitude and latitude $\pm 0.5°$. Figure 1 shows latitudinal time series of the AIRS and HIRDLS background variances during the measurement period between 2005 and 2008 at 42 km altitude. The overall structure in both data sets is rather similar. An annual cycle at high latitudes is detected which has during wintertime its maxima and during summertime its minima. The maximum in both data sets is up to $270\,K^2$ around $50°$ to $60°$

N/S. The activity of planetary waves is weaker in the southern hemisphere winter and in the southern hemisphere the polar vortex is more invariant in contrast to the northern hemisphere (e.g., Day et al., 2011). This is represented by the background variances which are larger in northern hemisphere winter than in southern hemisphere winter.

## 2.3   Estimation of retrieval noise

Temperature variances are notably affected by noise if long time spans or large areas are analyzed. Therefore it is fundamental to

characterize retrieval noise. For AIRS the noise was estimated directly from the measurements using the method of Immerkær (1996), following the approach of Hoffmann et al. (2014). Immerkær (1996) presented a generic technique for noise estimation developed for image analysis. Individual noise estimates are obtained for each AIRS granule and each altitude. The temperature data is nested with a $3\times3$ pixel filter mask which eliminates image structures. The variance of the filtered data is calculated which gives an approximation of the noise. Note that it is possible with the method of Immerkær (1996) to misinterpret plane

waves with very short horizontal wavelengths as noise, because thin lines are recognized as noise. However, based on inspection of the data we concluded that this issue does not affect our analysis.

Figure 2 shows global mean noise estimates for the temperature measurements of AIRS and HIRDLS on individual days. The noise estimate for AIRS is about 1.0 K at 24 km altitude and increases to 2.2 K at 55 km altitude. Seasonal differences of 10 % are found, with lowest values in January and highest values in July. Noise profiles of April and October are similar and located in between. These direct noise estimates from the temperature data agree well with the estimated retrieval noise, which is about 1.4 to 2.1 K in the altitude range between 20 and 60 km (Hoffmann and Alexander, 2009). Gravity wave variances of





AIRS are analyzed by subtracting the squared noise estimate from the temperature variances. For HIRDLS both a measured and a predicted precision are provided. The predicted precision corresponds to the expected uncertainty of the retrievals based
on uncertainty of the input parameters. This includes the radiance noise, but also other parameters, e.g., forward model errors (Khosravi et al., 2009a, b; Gille et al., 2011). The theoretically estimated temperature precision of HIRDLS has no seasonal variability and is about 0.6 to 1.7 K, increasing with altitude (see Fig. 2). Additionally to this theoretical estimate, the precision can be estimated directly from the observed temperature profiles after the retrieval (Gille et al., 2011). This estimate, however, includes some of the effects of small-scale wave motions, especially gravity waves. This precision is about 0.3 K at 20 km
and increases to 0.6 K at 50 km. Noise was not corrected for in our HIRDLS analysis, because the values of the zonal average standard deviations, which are attributed to gravity waves, and the theoretically expected precision are larger.

## 2.4 Sensitivity functions of AIRS and HIRDLS

Each type of current satellite instruments can detect only a certain part of the full vertical and horizontal wave number spectrum of gravity waves, which is determined by its observational filter (Alexander, 1998; Preusse et al., 2008; Alexander et al., 2010;
Trinh et al., 2015). For AIRS the sensitivity to vertical wavelengths was determined following Hoffmann et al. (2014), i.e., vertical temperature profiles, which represent wave perturbations are convoluted with the averaging kernel functions. The variance of the resulting temperature perturbations for all wave phases was related to their overall maximum. For waves whose amplitude is constant with height the sensitivity was determined. Therefore it was for horizontal wavelengths the detrending procedure on wave packages in the across-track direction applied and the ratio of the variances of the detrended perturbations
for different wave phases was calculated to their overall maximum.

The sensitivity function of limb sounders is really two dimensional and the sensitivity for horizontal and vertical wavelengths can not be estimated independently. The calculation of the HIRDLS sensitivity function follows the approach of Preusse et al. (2002), with additional vertical filtering being applied. This additional filtering was added because in the analysis by Ern et al. (2011) gravity wave amplitudes are determined in sliding windows of 10 km vertical extent. Amplitudes with vertical wave-
lengths longer than 25 km can not be reliably determined from those windows and therefore only vertical wavelengths up to 25 km are used in the vertical analysis of altitude profiles. This vertical analysis is a two-step approach utilizing the maximum entropy method for identifying the dominant vertical oscillations, followed by a harmonic analysis (MEM/HA). For more details see Preusse et al. (2002). As second aspect the vertical filtering will further reduce contamination by planetary waves in the polar vortex. These waves usually have long vertical wavelengths of around 40 km and longer.
Figure 3 illustrates the sensitivity functions for AIRS and HIRDLS for gravity wave temperature variances. Only waves with horizontal wavelength longer than 20 km can propagate from the troposphere into the stratosphere (Preusse et al., 2008), therefore the horizontal wavelength in the plots are cut below 20 km. The sensitivity of AIRS exceeds the 20% level for vertical wavelengths longer than 15 km and horizontal wavelengths shorter than 1280 km. Highest sensitivity is found for long vertical and short horizontal wavelengths, as expected for a nadir sounder. In contrast, the observational filter of HIRDLS shows the typical picture for limb sounders with high sensitivity for short vertical and long horizontal wavelengths. The 20% level of sensitivity is exceeded for vertical wavelengths longer than 2 km and shorter than 39 km and for horizontal wavelengths longer





than 140 km. The horizontal wavelengths considered in the HIRDLS sensitivity function are the wavelengths along the line-of-sight of the satellite. The true wavelength is usually shorter than this projection. Therefore limb scanners can detect gravity

waves with even shorter horizontal wavelength than suggested by the sensitivity function. Assuming that horizontal wave vectors of observed gravity waves are randomly distributed, the average horizontal wavenumber would be underestimated by a factor of $\sqrt{2}$, giving a rough measure of how much shorter observed true horizontal wavelengths could be on average.

Supposing the same relative potential temperature amplitudes for two waves with different values of horizontal and vertical wavelengths, waves with short horizontal and long vertical wavelength can potentially carry more gravity wave momentum

flux. We calculated a momentum flux factor $M(k_h, m)$, which gives a rough estimate how much waves of different horizontal and vertical wavenumbers $k_h$ and $m$ could possibly contribute to momentum flux,

$$F_{ph} = M(k_h, m) \times \left( \frac{\hat{T}}{T} \right)^2, \tag{2}$$

for a given normalized wave amplitude $\hat{T}/T$. Following Ern et al. (2004), the momentum flux factor is calculated according to

$$M(k_h, m) = \frac{1}{2} \rho \left( \frac{g}{N} \right)^2 \frac{k_h}{m} A B, \tag{3}$$

$$A = \left[ 1 - \frac{\hat{\omega}^2}{N^2} \right] \times \left[ 1 + \frac{1}{m^2} \left( \frac{1}{2H} - \frac{g}{c_s^2} \right)^2 \right]^{-1} \times \left[ 1 + \left( \frac{f}{m\hat{\omega}} \right)^2 \left( \frac{1}{2H} - \frac{g}{c_s^2} \right)^2 \right]^{1/2}, \tag{4}$$

$$B = \left| \left( \hat{\Theta}/\bar{\Theta} \right)^2 / \left( \hat{T}/\bar{T} \right)^2 \right|. \tag{5}$$

with density $\rho$, gravity acceleration $g$, buoyancy frequency $N$, intrinsic frequency $\hat{\omega}$, scale height $H$, sound speed $c_s$, Coriolis

parameter $f$, and potential temperature $\Theta$. The black contour lines shown in both panels of Fig. 3 indicate the normalized momentum flux factor, $M'(k_h, m) = M(k_h, m)/M_{max}$, which is normalized by the maximum value $M_{max}$ that occurs in the horizontal and vertical wavelengths range shown. The normalized momentum flux factor can attain values between near 0 and 1. Of course the normalized momentum flux factor is just a scaling factor that does not provide information about the relative occurrence rate of waves with given horizontal and vertical wavelengths in the atmosphere. Here we give an example of the

importance of the momentum flux factor in interpreting the AIRS and HIRDLS gravity wave observations. Assuming that HIRDLS observes a gravity wave with 600 km horizontal wavelength and 6 km vertical wavelength (which is well within its sensitivity range), the corresponding normalized momentum flux factor is 0.02. Further, assuming that AIRS observes a gravity wave with 200 km horizontal wavelength and 30 km vertical wavelength, the corresponding normalized momentum flux factor is 0.26. The gravity wave observed by AIRS would contribute a factor 10 more momentum flux than HIRDLS, if both had the same amplitude.



## 3 Comparison of AIRS and HIRDLS gravity wave observations

### 3.1 Case studies of individual wave events

Following Hoffmann and Alexander (2009), in this section individual gravity wave events in the AIRS data are compared with HIRDLS observations at the same location and at a similar time. Overpass times of the same geographic locations are for AIRS and HIRDLS within minutes, because both are member of the A-Train constellation of NASA satellites. Based on their different viewing geometries, AIRS as nadir sounder and HIRDLS as limb sounder with fixed azimuth angle of -47°, the times where AIRS and HIRDLS see the same geographic locations differ by about 100 min. The gravity wave patterns can change substantially on timescales of 100 min, in particular in case of gravity waves with high frequencies and fast group velocities. Variations in the phase structure of mountain waves are more likely invariant in a 100 min interval in contrast to waves from other sources, because they are stationary relative to the ground. Mountain waves are therefore best suited for a direct comparison of AIRS and HIRDLS data. However, we analyzed several gravity wave events of different sources, which are observed by both AIRS and HIRDLS. Figures 4 and 6 show temperature perturbation maps of the AIRS operational retrieval and the AIRS high-resolution retrieval, as well as HIRDLS measurement locations at 30 and 42 km altitude. In Figs. 5 and 9 the corresponding vertical cross-sections of the AIRS operational retrieval, the AIRS high-resolution retrieval, and HIRDLS are presented. The AIRS measurements have been linearly interpolated to the HIRDLS track for this comparison.

The first case shows a mountain wave event at Tierra del Fuego, South America, on 29 September 2006 (Figs. 4 and 5). This case was also investigated by Hoffmann and Alexander (2009), but a different analysis of the HIRDLS data is used in this study. The results found by Hoffmann and Alexander (2009) are reproduced successfully. The vertical maps and cross-sections of the temperature perturbations from the AIRS high-resolution retrieval and HIRDLS agree well in amplitude and phase structure of the mountain wave event. Remaining differences are likely due to the different vertical resolution of both instruments. Note that the AIRS operational retrieval also shows this event, but the retrieved wave amplitudes are significantly lower. The vertical resolution of the operational retrieval is also significantly degraded compared with the high-resolution retrieval above 40–45 km. This is attributed to stronger smoothing constraints in the operational retrieval.

The second case study shows a non-orographic wave event over the southern Indian Ocean on 8 August 2007 (Figs. 6 and 9), which was likely initiated by jet or storm sources. Figure 7 shows in the upper panel a zonal average of the horizontal wind of ERA-Interim and in the lower panel the horizontal winds at 243 hPa (about 10 km) and 13.9 hPa (about 30 km). In the zonal average of the horizontal wind the jets at the upper troposphere lower stratosphere and in the polar stratosphere are clearly seen. The maps at 243 hPa and 13.9 hPa show the polar front jet, too. The exit region of the jets, where gravity wave generation is common, is located at the position of the wave event. Figure 8 shows 8.1 $\mu$m brightness temperatures of AIRS. This map indicates the presence of a storm system, which could also be a source for the gravity wave event. The temperature perturbation maps show that the HIRDLS track is at the edge and catches mostly the western part of the wave event. Nevertheless, the vertical cross-sections of the AIRS high-resolution and HIRDLS retrievals show a similar structure, with larger amplitudes in HIRDLS and slightly larger vertical wavelengths in AIRS. The coarser vertical resolution of AIRS is obvious in the vertical cross-section and results in an attenuation of the amplitudes and coarser vertical structures compared to HIRDLS. This effect increases with





altitude, which can be attributed to decreasing vertical resolution of the AIRS retrieval with height. A comparison between the AIRS operational and high-resolution retrieval shows a severe attenuation of the amplitude of the wave event and the coarser horizontal resolution of the operational data. These case studies illustrate that despite the rather different sensitivity functions AIRS and HIRDLS are capable of observing gravity waves from the same sources in individual events.

## 3.2 Time series of gravity wave variances

This section focuses on time series of gravity wave variance of AIRS and HIRDLS at about 30 km and 42 km altitude during January 2005 to March 2008. The temporal development and latitudinal structure of the gravity wave variance at 30 km is shown in Fig. 10 and at 42 km in Fig. 11. A detailed picture for four selected latitudes at 42 km is given by Fig. 12. Additionally, in all figures the zonal mean wind of ERA-Interim at the chosen altitude is shown. Latitudes 44°N and 47°S in Fig. 12 are chosen, because they are the maximum and minimum latitudes, which are completely covered by AIRS measurements. We found that the seasonal cycle is captured very well in the AIRS and HIRDLS data sets and the structure is rather similar. Apart from the wintertime maxima in the polar regions, gravity wave variance between 50°S and 50°N is usually between 0.1 and 0.5 $K^2$ (30 km) and 0.5 and 2 $K^2$ (42 km) for AIRS high-resolution retrieval and between 1 and 2 $K^2$ (30 km) and 2 and 5 $K^2$ (42 km) for HIRDLS. In the subtropics a weaker annual cycle with maxima during summertime and minima during wintertime is found. These summertime maxima have been observed before (e.g. Jiang et al., 2004b; Ern and Preusse, 2012; Hoffmann et al., 2014), and they have been attributed to stronger activity of deep convective sources during summer (e.g. Choi et al., 2012; Trinh et al., 2016). Additionally, a major effect is the modulation of wave amplitudes by the background winds. We found an annual cycle at high latitudes, which has during wintertime its maxima and during summertime its minima. The highest values are found at the polar vortex in the southern hemisphere with values up to 9 $K^2$ for AIRS high-resolution retrieval and up to 29 $K^2$ for HIRDLS. During boreal wintertime 2007 a double-peaked maximum at 44°N is seen in AIRS high-resolution retrieval and HIRDLS. AIRS high-resolution retrievals detected a double-peaked maximum during boreal wintertime 2006 at 44°N, which is not seen in HIRDLS at this latitude but somewhat further north. In January 2006 a major sudden stratospheric warming (SSW) occurred and the double peak structure is likely related to the SSW. In the high-resolution retrieval of AIRS it could be seen, with a small delay, that the gravity wave activity is strengthening after the SSW when the zonal wind increases again. For an overview of gravity wave activity in the northern hemisphere polar region during recent winters see Ern et al. (2016). Hoffmann et al. (2016) discussed gravity wave activity located at southern hemisphere orographic hotspots and their correlation with background winds in more detail.

Comparing zonal winds at 2.5 hPa (about 42 km) and stratospheric gravity wave variances a strong correlation can be found for both AIRS and HIRDLS. The largest gravity wave variances occur in mid- to high-latitude regions where stratospheric zonal mean winds are $\sim 25 \, \mathrm{m \, s^{-1}}$ or greater. At 44°N and 47°S the maxima during wintertime correspond with strong westerly zonal winds, up to $110 \, \mathrm{m \, s^{-1}}$ at 47°S. At 20°N and 20°S maxima during summertime match well with strong easterly zonal winds. It is often observed that gravity wave activity is amplified in the presence of strong background winds (e.g., Wu and Waters, 1996a, b; Jiang and Wu, 2001). If the phase speeds of gravity waves are opposite to the background wind their saturation amplitudes are enlarged. An additional effect is that the vertical wavelength of these gravity waves is Doppler shifted towards





longer vertical wavelengths, which are better visible in particular for AIRS. A more detailed discussion of this effect can be found, for example, in Ern et al. (2015) and Hoffmann et al. (2016). This also means that long vertical wavelength gravity waves are preferentially found in regions of strong background winds. This is the likely reason why in Fig. 11 the patterns of
AIRS gravity wave variances match the distribution of the background winds somewhat better than the HIRDLS variances.

The values of the operational retrieval are a factor of two lower if it they are compared to the AIRS high-resolution retrieval. At 44°N no double peak related to the SSW is seen in AIRS operational retrieval values during boreal wintertime 2006 and 2007. At 20°N and 20°S gravity wave variances during wintertime are not increasing, which is seen in both AIRS high-resolution retrieval and in HIRDLS. Obviously, the AIRS high-resolution retrieval is more suitable for the analysis of gravity
waves than the AIRS operational retrieval due to the better horizontal resolution and improved vertical resolution.

### 3.3  Influence of sensitivity functions on gravity wave variances

As we conducted a full spectral analysis of the HIRDLS data, we are able to apply the AIRS sensitivity functions to the HIRDLS data in order to estimate the fraction of variances that is actually observed by both instruments. For this procedure horizontal and vertical wavelengths of the gravity waves are required. From the HIRDLS measurement track consecutive altitude profiles,
which observe the same gravity wave, are used to determine horizontal wavelengths. This approach has been used before to estimate gravity wave momentum fluxes from satellite data (e.g., Ern et al., 2004). The average sampling distance between these consecutive altitude profiles is 90 km, and the profiles are observed within only about 15 sec. Therefore often the same gravity wave should be observed in consecutive profiles, and due to the short sampling times the wave field should not change due to the oscillation frequency of the wave. The horizontal structure of the wave is responsible for phase differences. Nevertheless,
to ensure that in successive profiles the same gravity wave is looked at, only waves with the vertical wavelengths differing by no more than 40 % in the two profiles of a pair are selected. The fraction of selected pairs with respect to the total number of possible pairs is thereby reduced to about 60–70 % at low latitudes, and to about 50–60 % at high latitudes. Gravity wave variances due to the strongest gravity wave components in all single profiles without pair selection and of the selected pairs are almost exactly the same. Therefore the selected pairs are considered to be representative for the global distribution of all
gravity waves. However, there will always be an angle $\alpha$ between the horizontal wave vector of the gravity waves $\mathbf{k_{GW}}$ and the sampling track of the satellite. The observed horizontal wavenumber $k_{obs}$ will therefore underestimate $\mathbf{k_{GW}}$ by a factor $cos(\alpha)$, and the horizontal wavelength will be overestimated by a factor $1/cos(\alpha)$.

Figure 13 illustrates the influence of the observational filter of AIRS to the HIRDLS gravity wave variances by showing HIRDLS gravity wave variances with and without the AIRS observational filter being applied. Additionally, gravity wave
variances of the AIRS high-resolution retrieval are shown. Plotted are time series of the gravity wave variance at 42 km altitude for the same latitudes as in Sect. 3.2 from HIRDLS, HIRDLS with MEM/HA, AIRS high-resolution retrieval and HIRDLS filtered with AIRS sensitivity function. Note that for a better identification the results from HIRDLS filtered data sets were scaled by a factor of 5. The HIRDLS gravity wave variance is significantly reduced after the AIRS observational filter is applied. HIRDLS filtered with AIRS sensitivity function reproduces at the maximum 8 % at 47°S and at the minimum 3 % at 20°N of the HIRDLS gravity wave variance. Values of HIRDLS including the AIRS observational filter are considerably




lower than values directly from the AIRS high-resolution retrieval. This confirms that there is only small spectral overlap of the HIRDLS and AIRS sensitivity functions and points to an under-representation of small horizontal-scale waves in HIRDLS data compared with AIRS. Still, relative variations are very similar and some structures seen in AIRS became visible in HIRDLS

gravity wave variances after including AIRS observational filter. At 44°N the filtered HIRDLS gravity wave variances show the double peak structure during boreal wintertime 2006, which is not seen in unfiltered data. The gravity wave activity is strengthening after the SSW when the zonal wind increases again in both filtered HIRDLS gravity wave variances. This is also seen in AIRS, but somewhat delayed. During boreal winter 2006 and 2007 the filtered HIRDLS gravity wave variances are more gradually decreasing with time at 44°N after the peak value than in the unfiltered HIRDLS gravity wave variances.

This behaviour is very similar as in the AIRS gravity wave variances. The analysis confirms that AIRS and HIRDLS gravity wave measurements can be considered complementary to each other, because they observe diverse sections of the gravity wave spectrum. The relative variations in all time series are similar, which indicates that these variations are induced by similar physical processes (e.g., wind effects and source mechanisms). Therefore it might be possible to transfer directional information obtained for AIRS to HIRDLS observations.

## 4    Summary and conclusions

In this study we compared temperature variances of AIRS and HIRDLS to evaluate the relationship of their stratospheric gravity wave observations. Our analyses are performed on the HIRDLS operational retrievals, AIRS operational retrievals, and a dedicated AIRS high-resolution data set. The measurement geometries of AIRS (nadir) and HIRDLS (limb) are diverse and therefore they have opposite sensitivities to horizontal and vertical wavelengths, which is shown by their sensitivity functions.

However, a comparison of individual orographic and non-orographic gravity wave events showed that stratospheric wave structures of AIRS and HIRDLS agree very well, which is consistent with earlier work of Hoffmann and Alexander (2009). With respect to the AIRS high-resolution retrievals, the case studies demonstrate that AIRS and HIRDLS agree generally well in amplitude and phase structure for a mountain wave event and a non-orographic wave event. AIRS has coarser vertical resolution, which results in an attenuation of the amplitude and coarser vertical structures compared to HIRDLS, which is much

more evident for the AIRS operational retrieval. However, AIRS has a much higher horizontal resolution and the propagation direction of the wave can be clearly identified in geographical maps of the wave events. The horizontal orientation of the phase fronts can be deduced from AIRS 3D temperature fields. This is a restricting factor for gravity wave analyses of limb measurements.

A comparison of time series of gravity wave variance of AIRS and HIRDLS revealed that HIRDLS gravity wave variances

show an offset due to regular background activity of gravity waves and are typically about a factor of 3–5 larger than for AIRS. This is attributed to the different measurement geometries and the limitation to long vertical wavelengths for AIRS in particular. We calculated a momentum flux factor, which gives a rough estimate how much the waves given horizontal and vertical wavelengths and amplitude contribute to momentum flux, if they exist in the real atmosphere. It indicates that the waves with short horizontal and long vertical wavelengths seen by AIRS contribute significantly to momentum flux, even





if the AIRS temperature variance may be small compared to HIRDLS. Despite this systematic difference, the seasonal and latitudinal distributions of stratospheric gravity wave activity found in both data sets are rather similar. Overall, these variations are related to the well-known seasonal patterns of gravity wave activity with summertime maxima in the subtropics, and

wintertime maxima at high latitudes (e.g., Ern et al., 2011, 2013; Hoffmann et al., 2013, 2014). Several sources of gravity waves can produce these maxima. Because of the stronger activity of deep convective sources during summer, the summertime maxima in the subtropics occur. Gravity wave variances show great enhancement in the winter hemisphere over mid and high latitudes where the polar night jet is strongest (Plougonven and Zhang, 2014) and due to strong mountain wave activity (Jiang et al., 2004a). The seasonal distribution of stratospheric gravity wave activity found in this study agrees well with other

satellite climatologies based on limb measurements (e.g., Preusse et al., 2009a). The gravity wave variances agree qualitatively well with the AIRS climatology of Gong et al. (2012), which is based on 15 $\mu$m radiance measurements and of Hoffmann et al. (2013), which is based on 4.3 $\mu$m brightness temperature variances.

    Wright et al. (2011) compared HIRDLS, COSMIC, and SABER detections of stratospheric gravity waves during the years 2006–2007 and concluded that, when allowing for their different vertical resolution capabilities, the three instruments repro-

duce each others results for magnitude and vertical scale of perturbations to within their resolution limits in approximately 50 % of cases. In a second study Wright et al. (2016) investigated, if the dissimilar results of many gravity wave studies are primarily of instrumental or methodological origin. Their analysis is located around the southern Andes and Drake Passage with different gravity wave resolving instruments. Their results show important similarities and differences. Limb sounder measurements show high intercorrelation between any instrument pair. AIRS and radiosonde observations tend to be uncorre-

lated or anticorrelated with the other data sets, suggesting very different behaviour of the wave field in the different spectral regimes accessed by each instrument. Evidence of wave dissipation is seen and varies strongly with season. In contrast to these two studies, we focus on a global statistical comparison of a nadir instrument (AIRS) and a limb instrument (HIRDLS) over a measurement period of three years. The data sets of AIRS and HIRDLS are found to be complementary to each other. AIRS primarily observes only the short horizontal and long vertical wavelength waves and HIRDLS primarily observes only the long

horizontal and short vertical wavelength waves. To address the differences between the AIRS and HIRDLS distribution to the different sensitivity functions a simple approach of filtering HIRDLS data with the AIRS sensitivity function was conducted. Still, relative variations are very similar and some structures seen in AIRS became visible in HIRDLS gravity wave variances after including the AIRS sensitivity function. Of course, not all differences can be explained by this simple approach, but it might be possible to transfer directional information obtained for AIRS to HIRDLS observations for case studies.

    In summary, despite the different sensitivity function, AIRS and HIRDLS are capable of observing gravity waves from the same sources in individual events, and their relative distributions of gravity wave variances agree well. The analysis confirms that AIRS and HIRDLS observe largely different sections of the gravity wave spectrum, but they complement each other and thereby larger parts of the gravity wave spectrum can be observed. Combining the observations would be a great chance for gravity wave research in the future.





*Data availability.* AIRS and HIRDLS data are distributed by the NASA Goddard Earth Sciences Data Information and Services Center (GES DISC). ERA-Interim data were obtained from the European Centre for Medium-Range Weather Forecasts (ECMWF).

*Author contributions.* All authors contributed to the design of the study and provided input to the manuscript. The data for the study was processed by CIM and additionally she produced all figures, and drafted the text. The 3D AIRS retrieval scheme was developed and the used 3D AIRS data are produced by LH and MJA. ME produced the HIRDLS data set used. QTT provided the HIRDLS observational filter data.

*Competing interests.* The authors declare that they have no conflict of interest.

*Acknowledgements.* The work by M. Ern was partly supported by Deutsche Forschungsgemeinschaft (DFG) grant no. PR 919/4-1 (MS-GWaves/SV).





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





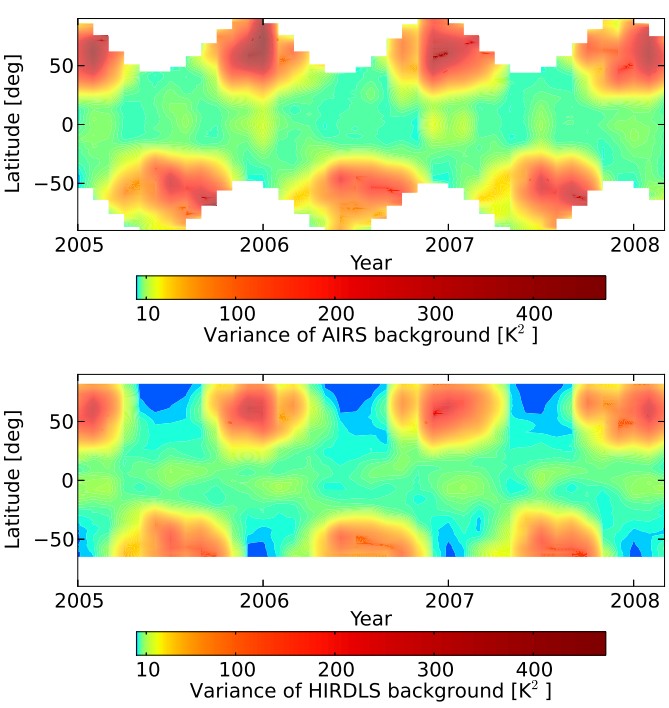

**Figure 1.** Time series of monthly mean temperature background variances for measurements between 2005 and 2008 at 42 km altitude. Top: AIRS high-resolution retrieval. Bottom: HIRDLS operational retrieval. Data gaps in AIRS data (white areas) are related to the restriction to nighttime measurements.



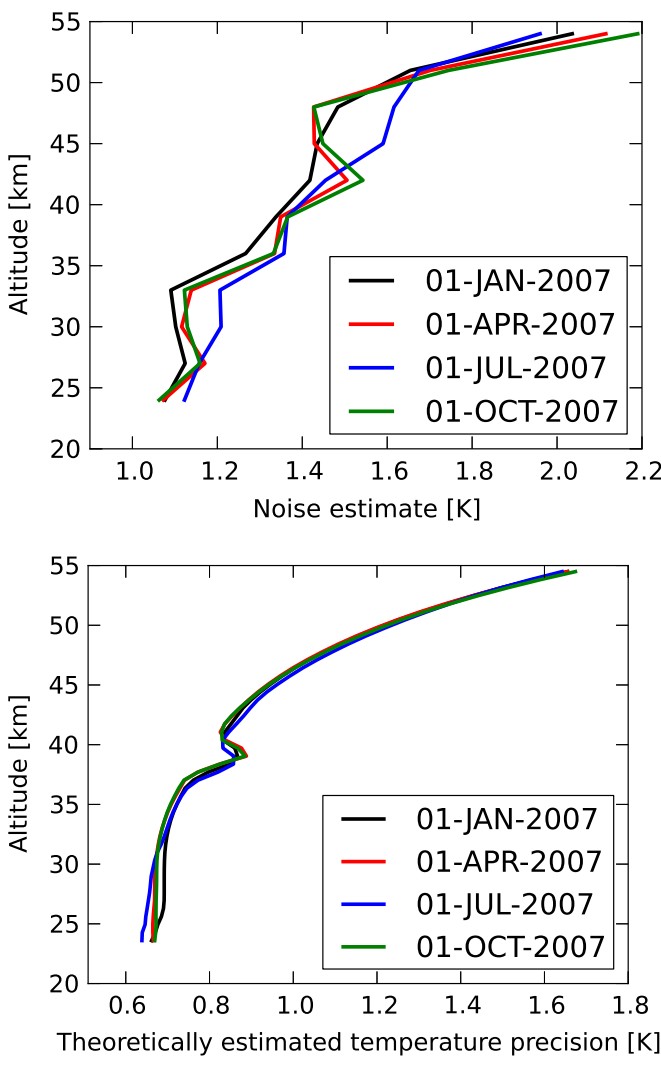

**Figure 2.** Estimated global mean noise profiles for AIRS (top) and HIRDLS (bottom).



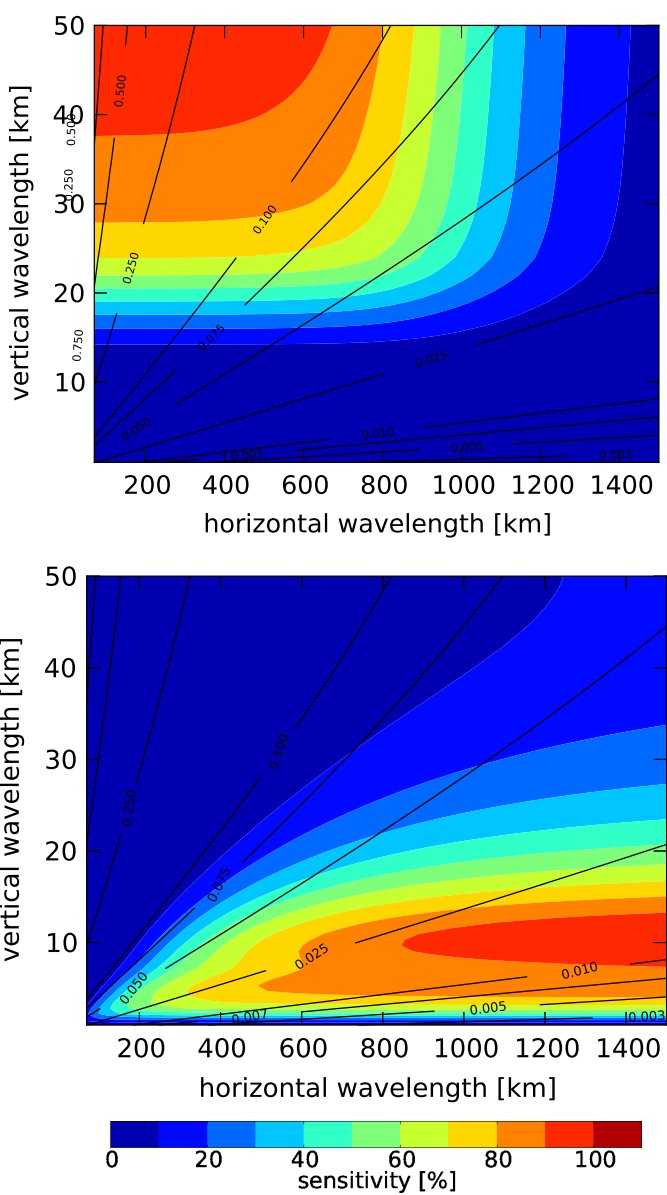

**Figure 3.** AIRS (top) and HIRDLS (bottom) observational filters indicate the sensitivity of temperature variances to gravity waves with different horizontal and vertical wavelengths. The black lines show a momentum flux factor (see text for details).




**Figure 4.** Temperature perturbations from AIRS retrievals on 29 September 2006 about 3 UTC at 30 km (left) and 42 km (right) for a mountain wave event near Tierra del Fuego.. Top: AIRS operational retrieval. Bottom: AIRS high-resolution retrieval. Black circles indicate the locations of HIRDLS profiles.



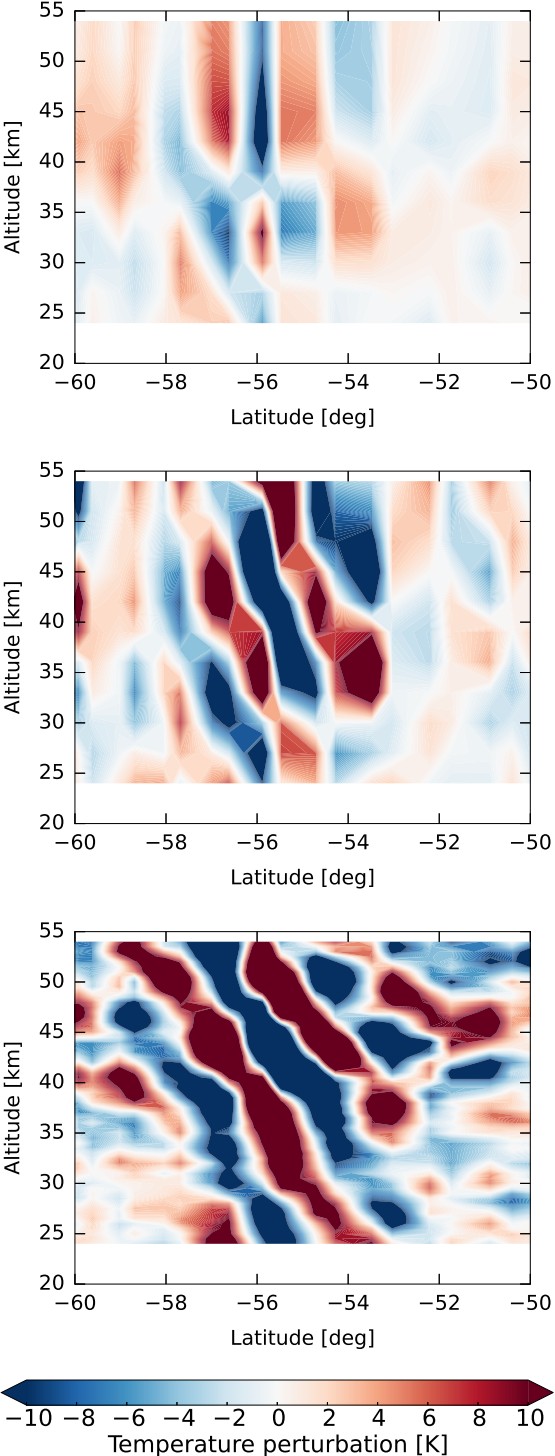

**Figure 5.** Vertical cross-sections of temperature perturbations on 29 September 2006 about 3 UTC for a mountain wave event derived from the AIRS operational retrieval (top), the AIRS high-resolution retrieval (middle), and HIRDLS (bottom).





**Figure 6.** Same as Fig. 4, but for a non-orographic gravity wave event over the southern Indian Ocean on 8 August 2007, about 17 UTC.



**Figure 7.** Top: Zonal average of horizontal wind of ERA-Interim for a non-orographic gravity wave event over the southern Indian Ocean on 8 August 2007, 18:00 UTC. Bottom: Horizontal wind maps of ERA-Interim.





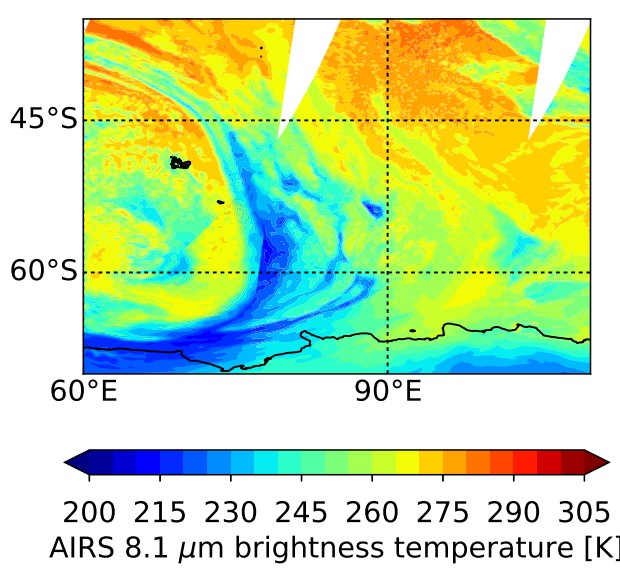

**Figure 8.** 8.1$\mu$m brightness temperature measurements of AIRS for a non-orographic gravity wave event over the southern Indian Ocean on
8 August 2007. Low brightness temperatures indicate the presence of a storm system in the study area.





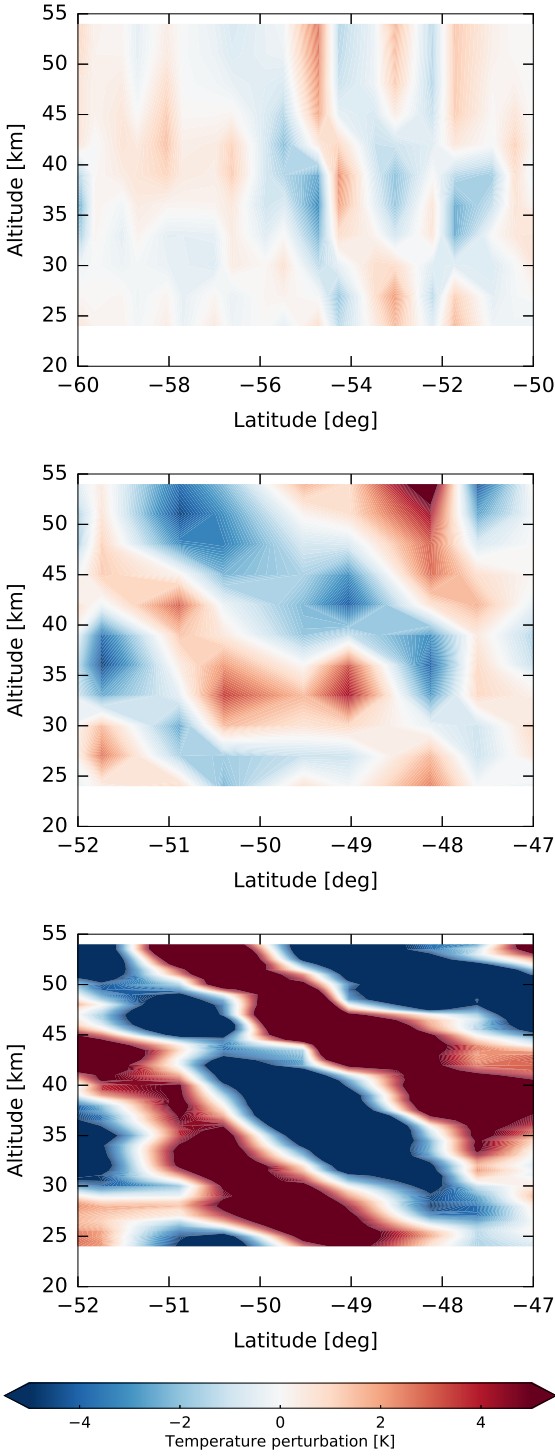

**Figure 9.** Same as Fig. 5, but for a non-orographic gravity wave event over the southern Indian Ocean on 8 August 2007, about 17 UTC.



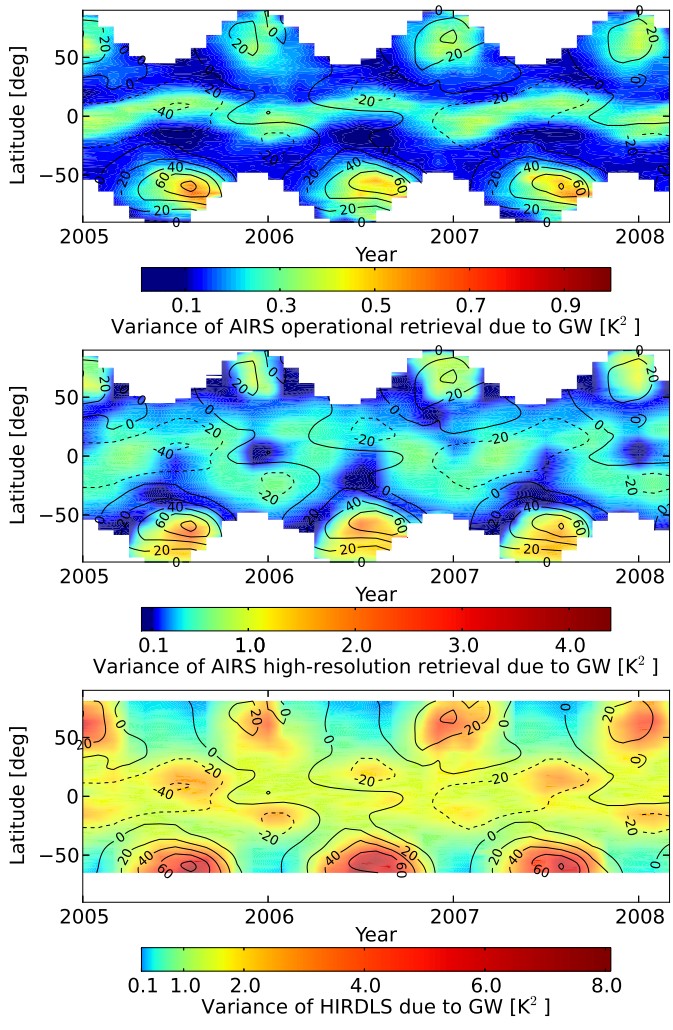

**Figure 10.** Time series of monthly temperature variances due to gravity waves between 2005 and 2008 at 30 km altitude. Top: AIRS operational retrieval. Middle: AIRS high-resolution retrieval. Bottom: HIRDLS. Contour lines indicate zonal mean wind from ERA-Interim. Please note the different color bar ranges.



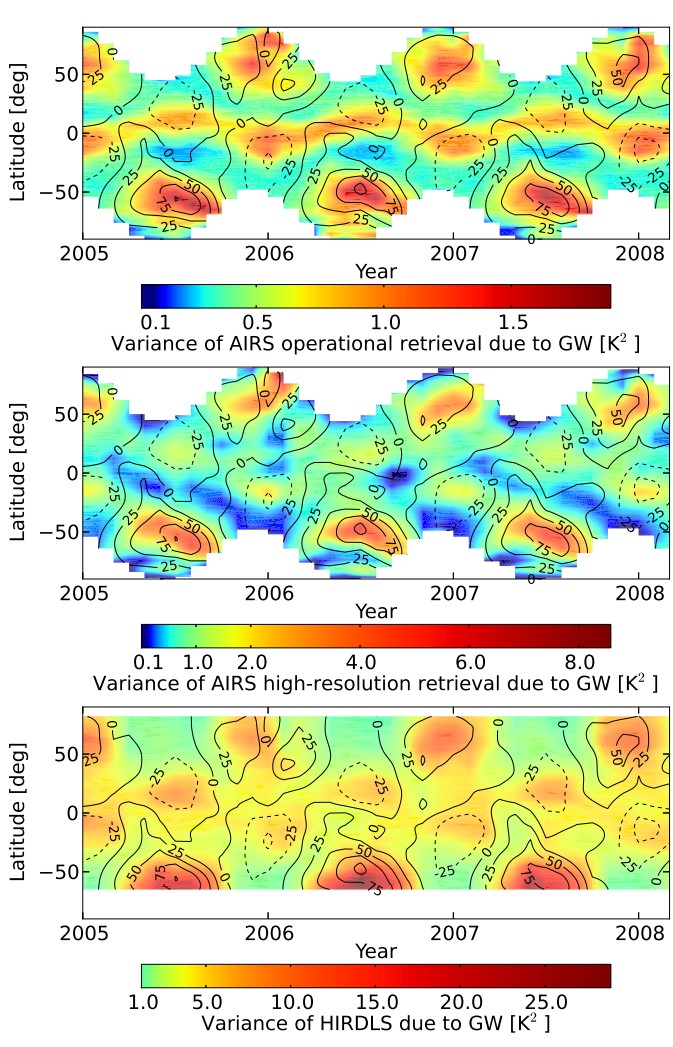

**Figure 11.** Same as Fig. 10 but for 42 km.





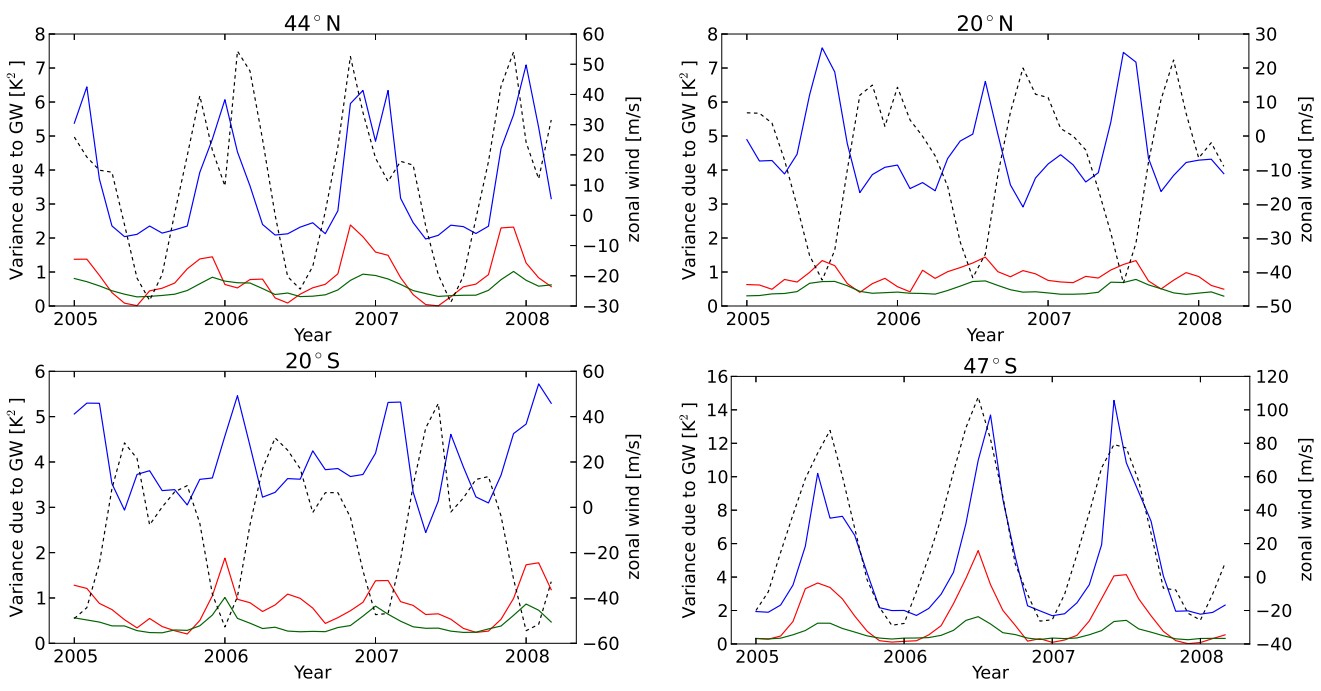

**Figure 12.** Time series of monthly mean gravity wave variances for measurements between 2005 and 2008 at 42 km altitude and different latitudes (see plot titles). Green: AIRS operational retrieval. Red: AIRS high-resolution retrieval. Blue: HIRDLS. Black dashed lines indicate zonal mean winds at 2.5 hPa from ERA-Interim.





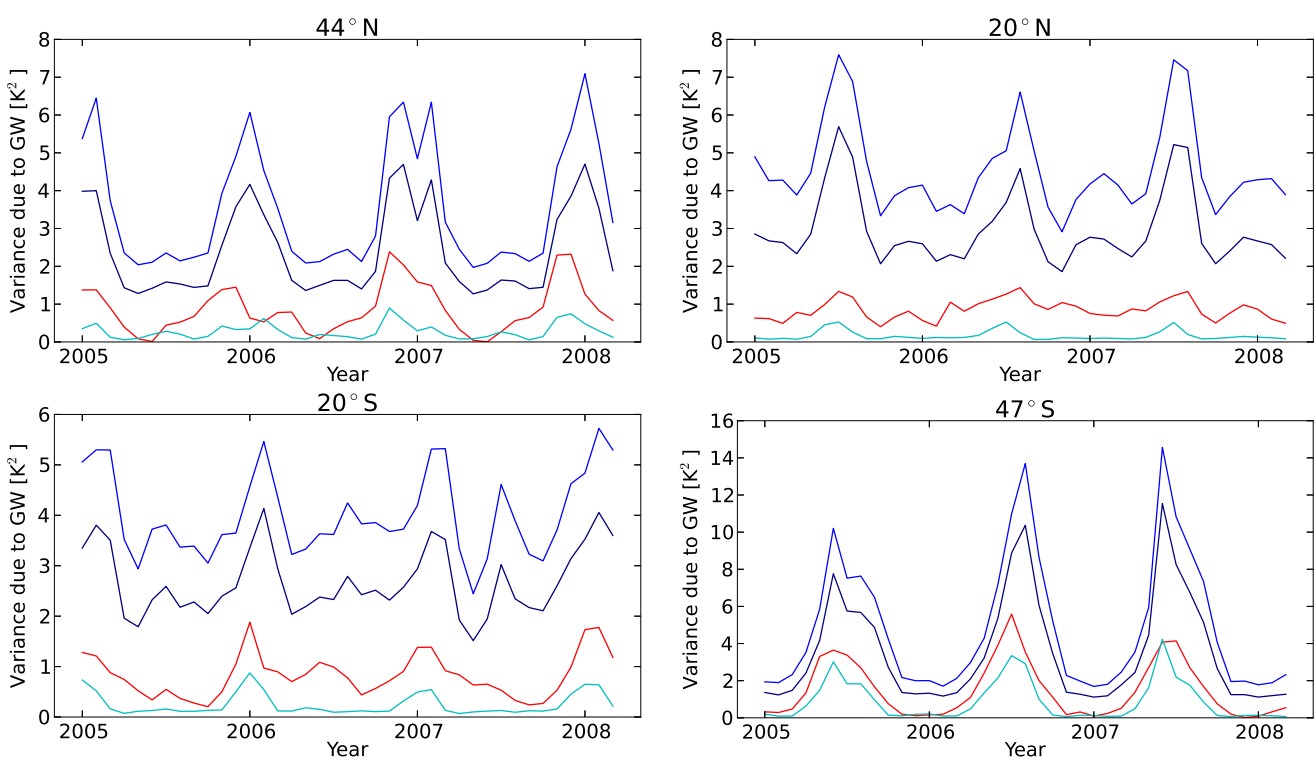

**Figure 13.** Time series of gravity wave variances at 42 km altitude and different latitudes (see plot titles). Red: AIRS high-resolution retrieval. Light blue: HIRDLS. Dark blue: HIRDLS with MEM/HA. Cyan: HIRDLS filtered with AIRS sensitivity function. Note that filtered HIRDLS data are scaled by a factor of 5.