# Peer review of "Intercomparison of AIRS and HIRDLS stratospheric gravity wave observations"

_Atmospheric Measurement Techniques, 2017_

## Referee Comment (RC1) · M.nbsp;A. Geller (Referee) · 3 Sep 2017

This is an excellent paper. The authors have used both AIRS and HIRDLS observations to study atmospheric gravity waves. AIRS is a nadir-viewing instrument, and hence has relatively poor altitude resolution, but AIRS, having cross-track scanning capability, has excellent horizontal resolution. HIRDLS is a limb-viewing instrument that has better vertical resolution, but due to a malfunction has fixed azimuth viewing. Although the two spacecraft on which these instruments are flying, Aqua in the case of AIRS and Aura in the case of HIRDLS have overpasses separated by only a few minutes. The time separation between observations at the same point is actually about 100 minutes, a time separation over which gravity waves can vary considerably, so cases investigated in this paper have been chosen to hopefully minimize the influence of this.

[Figure]

Another point made in these papers is that the high-resolution AIRS retrievals are superior to the operational retrievals for measuring gravity wave variances. The operational retrieval uses 3 x 3 observational points. This is done to improve retrievals in the presence of clouds, but this is mainly important for the troposphere. The authors show that their high-resolution AIRS retrievals, which use each individual viewing point give superior stratospheric gravity wave information relative to the operational retrievals.

The measure of gravity wave activity used in this paper is gravity wave variance, but to obtain this, the variances due to larger scale atmospheric variability plus the variance due to instrumental noise must be subtracted from the measured variance. This is discussed in considerable detail in the early portions of the paper.

Now, the lower altitude resolution and higher horizontal resolution of AIRS relative to HIRDLS means that higher frequency gravity waves will preferentially be seen by AIRS relative to HIRDLS. A point made both early and later in the paper is that these higher frequency waves, with shorter horizontal and longer vertical wavelengths, will carry more momentum than the lower frequency waves seen by HIRDLS, even if the variances seen by the two are similar.

The gravity wave variances seen by AIRS and HIRDLS are compared for two cases. The first is for a mountain wave event, and the second is a storm event with active moist convection. For both cases, it is illustrated that the high-resolution AIRS product is superior for sensing gravity wave variances relative to its operational counterpart, and also that the general distribution of gravity wave variances, in both the horizontal and vertical, from the high-resolution AIRS data closely resembles those of HRDLS, when one takes into account the different frequencies and wavelength sensitivities of AIRS and HRDLS. This certainly suggests the broad-spectrum source nature for gravity waves for both events.

Gravity wave variances at 2.5 hPa (about 42 km) show a large correlation with zonal winds at that level for both AIRS and HRDLS. It is interesting that evidence of a similar
correlation between winds at 200 hPa and lower stratospheric gravity wave activity was noted by Wang and Geller (2003).

One important conclusion of this paper is that given the superior altitude and vertical scanning capability of HRDLS, which allows estimates of gravity wave momentum fluxes to be made, along with the superior horizontal information from AIRS that results from its horizontal scanning capability, use of the two data sets in a complementary manner should allow gravity wave propagation direction to be inferred by AIRS, and using this information would allow for more certain gravity wave momentum flux information to be derived from HRDLS. Of course, this relies on the broad-spectrum nature of the gravity wave fields emanating from significant gravity wave sources. Since short horizontal and long vertical wavelength gravity waves carry large momentum fluxes, perhaps clever combination of the two data sets can also be used to place more certain bounds on gravity wave momentum fluxes from various sources.

This is a very well written paper, with one exception, and that is the somewhat awkward use of English in a few instances. Of course, this is understandable given that only one of the authors is a native English speaker. One example of this is on line 12 on page 1, where the verbal use is "are conform." The term "are similar" would be preferable in my mind. This terminology is seen again on line 22 on the same page. A similarly awkward terminology is on line 18 of page 12, where the wording "are diverse" is used instead of the more preferable (to me) "are different."

I also have a couple of relatively minor points that I would like to see dealt with in this paper. One is a greater emphasis on the implication of broad-spectrum sources of atmospheric gravity waves. Another is on lines 15 and 16 of page 2, where they point out that satellite observations are only sensitive to a certain portion of the gravity wave spectrum. Of course, this is true for all observational techniques, a point made in Alexander et al. (2010). I also think the authors might spend a little time pointing out the different vertical phase tilts in the high-resolution AIRS and HRDLS variances in figure 5. This is likely due to the different propagation characteristics of the portion

of the gravity wave spectrum seen by the two instruments.

All in all, this is an excellent paper that will be a valuable addition to the literature on atmospheric gravity waves. It would be fine as is, but I think it could be improved a bit by their considering my suggestions.

References

Alexander, M. J., Geller. M., McLandress, C., Polavarapu, S., Preusse, P., Sassi, F., Eckermann, S., Ern, M., Hertzog, A., Kawatani, Y., Pulido, M., Shaw, T. A., Sigmond, M., Vincent, R., and Watanabe, S.: Recent developments in gravity-wave effects in climate models and the global distribution of gravity-wave momentum flux from observations and models, Q. J. Roy. Meteor. Soc., 136, 1103-1124, 2010.

Wang, L. and Geller, M. A.: Morphology of gravity-wave energy as observed from 4 years (1998-2001) of high vertical resolution U. S. radiosonde data, J. Geophys. Res., 108, doi:10.1029/2002JD002786, 2003.
* * *

---

## Referee Comment (RC2) · Anonymous Referee #2 · 9 Sep 2017

Overall comments:

The manuscript presents some interesting and new results on how well gravity wave results from HIRDLS and AIRS high-resolution retrievals agree with each other in statistical averages, and in some individual cases. It also presents informative results that extend and confirm previous conjectures on the complementarity of nadir and limb measurements, without, however, acknowledging some of that previous work sufficiently. The comparison of AIRS and HIRDLS observational filters is very nice, as are the comparisons of the two data sets for orographic and non-orographic waves, and the comparisons of seasonal patterns of variance. Although a minor point of the paper, the comparison of the gravity wave calculations based on AIRS operational and high-resolution data shows why the latter are needed.

[Figure]

However, the description of the instruments and data used is sometimes unclear, and occasionally wrong or misleading. Similarly, the description of the filtering is also occasionally unclear. The advantages of the filtering they have used, and the differences from alternative methods, is not spelled out.

The wording is sometimes poor or awkward.

Specific Comments:

Sec. 2.1 needs to be revised. The beginning is quite stilted. It could be noted that the 3x3 pattern of AIRS footprints fit within the footprint of the microwave instrument, which is used in the cloud-clearing approach. The discussion of the high- resolution data is needed, but should be made clearer. The source of the pressure mentioned on p. 4, l. 23 is not clear. Any additional references for the systematic errors and retrieval diagnostics would be useful if they exist. Do ll 35-36 mean that only nighttime data are used in this study? This seems to be the case, but t is not clearly stated. The range of the high-resolution retrieval is stated to be 10 to 70 km, with 5-6 degrees of freedom- does this mean that the vertical resolution is 10-12 km?

In the discussion of HIRDLS, it could be noted that HIRDLS was damaged during launch, precluding its planned ability to scan in azimuth, which would have given it 3D capabilities [Gille et al., 2003]. The damage resulted in its single view direction of -47° relative to the orbit plane. This also required extensive corrections to the processing algorithms [Gille et al., 2008, 2011]. Measurements of thermal emission with 1 km vertical resolution are made in 4 channels on the long-wave side of the 15 ïA■m bands, from which the temperature is retrieved as a function of pressure Khosravi et al. [2009a,b]. The Field of View of the instrument is always 1 km; the resolution of the retrieval varies with altitude.

Sec. 2.2: The "background removal" for AIRS is local, within one cross-track scan, ∼25°. It is noted that this strongly suppresses wave fronts parallel to the cross-track direction which cover large fractions of each scan. Why isn't this an important problem?
This seems much different from the method described for HIRDLS. Why couldn't this approach have been applied to comparable data from the overlapping 31 day time windows of HIRDLS data? It would be interesting to see how different those results would be from those used by Fetzer and Gille [1996], Alexander et al. [2008] and Wright et al. [2011, 2013], who used departures from 6 or 7 planetary scale waves that varied smoothly in time. Note that the HIRDLS V6 data also have a gridded product (using a Kalman Filter approach described in Gille et al., 2011). Please clarify the last sentence of the first paragraph on p. 6. It appears that all small-scale perturbations that get through the filtering are assumed to be GW. Is this correct? Is there evidence for this assumption? Please comment on last sentence of second paragraph: it is surprising that NH variance in winter is > SH variance in winter, given the large zonal winds and the Andes and Antarctic peninsula as such a large source.

Sec. 2.3: HIRDLS empirically estimated precisions for V6 appear to be underestimated. The values for V7 are much closer to the predicted precisions up to $\sim 0.5$ hPa, above which they are smaller [Gille et al., 2012]. The last sentence of this section is unclear.

Sec. 2.4: First paragraph- the treatment of wave phases is not clear. These sensitivity functions, and related discussion, are close to those of Wright et al. ACP 15, 8459-8477, 2015 [2015], which should be referenced and included in the discussion.

Sec. 4, toward end, could note that some combination of limb and nadir observations was done by Wright et al., GRL 43, 894, 2016.

Figures: Figure 8: Suggest second sentence change to . . .presence of high clouds associated with a storm system. . .

Technical comments:

p.1, l. 3: presumably vertical and horizontal resolution l. 12, also l. 22- better word than conform needed l. 18- better word than "fit" needed p.2, l 17 give overviews, l. 18

comparisons p. 3, l. 1 Suggest "Zonal average differences tend to be . . . l. 31 scan covers 1780 km p. 5, ll1,2: combine the first 2 sentences. l. 20: measurement typically consists p. 6, l. 28: data are p. 7, l. 16: perturbations, l. 21: The sensitivity function of the current generation of limb sounders. . .

p. 9, l. 31: Does the sentence beginning in this line refer to Figure 6? Text not clear.

p. 12, l. 28: . . .current limb measurements.
* * *

---

## Referee Comment (RC3) · Anonymous Referee #3 · 22 Sep 2017

The paper compares the gravity wave detection capabilities of the AIRS nadir sounder and the HIRDLS limb sounder. Reviewer 1 has already described the science area in some detail, so I will not repeat this except to say that the area is of significant current interest and the study is eminently suitable for AMT.

Reviewer 2 has already addressed several important technical details, and I agree with him/her that these should be addressed. In particular, I strongly agree with his/her comments that the large differences in background removal method are important, and will discuss this further in my comments below.

I also agree with both other reviewers that the language needs some work, although it is generally clear throughout and to a certain extent can be handled in copy-editing. Aside from this minor issue, the paper is well-structured and clear, and I suggest only

[Figure]

moderate additional revisions beyond those suggested by Reviewer 1 and 2.

=======Major comments=======

(specific references are listed at the bottom of the page)

1. I feel that the time difference between the two datasets could do with more consideration. This takes two main forms:

1a. in figures 4,5,6 and 9, the waves appear to be in almost exactly the same phase to the eye. For the mountain wave case, this is quite plausible; however, for the non-orographic case I'd like to see more evidence to confirm why this is so. In particular, since the full three-dimensional wavenumber vector can be inferred from the available data, it should be possible to infer the phase and group velocity of the wave (e.g. Fritts and Alexander 2003; Wright et al 2017), and hence confirm if the change between the two measurement times is indeed so small.

1b. in the global time series of variance, presumably there is a not-insignificant time-of-day difference between the two datasets. There's not much that can be done about this, but a little more discussion of how it may affect the results would be useful. This is likely to be most significant in the tropics, where convection has a diurnal cycle: while Aura and Aqua cross the equator in formation, the high viewing angle HIRDLS uses presumably means the scan track will cross at quite a different time than AIRS' nadir sensor.

2a. (also discussed by reviewer 2). The background-removal analysis is inconsistent between the two datasets. I'm not sure why this needs to be so: since global data is available for both AIRS and HIRDLS, presumably a common background removal method could be implemented, presumably more similar to the HIRDLS method used in the paper.

2b. also, why in particular is a fourth-order polynomial specifically used for the background removal? I realise this is in common with previous studies, but my understanding was that this was to remove solar glint from the AIRS radiances, which is presumably removed in the 3D temperature retrieval. [I am happy to be corrected on this!]

3. P10L21 onwards: You refer in passing to a double-peak in HIRDLS GW variance at 44N in winter 2007, with no attribution, but then explain in detail a similar features in the AIRS data as being due to an SSW. I definitely believe the AIRS feature - for example, the AIRS time series look extremely similar to figure 3 of Wright et al (2010) and it may be useful to say this - but it seems odd to focus in the text on this relatively small feature of the AIRS time series but not on the (to my eye) much larger change in the HIRDLS series in early 2007. Do you have any idea why the HIRDLS double-peak in early 2007 occurs?

4. The idea of combining limb and nadir sounders has been used previously, for example by Wright et al (GRL, 2016) [and references therein]. It would be useful to mention this in your conclusions, where you suggest that combining limb and nadir datasets for better coverage would be useful. [I realise there are important differences in the two approaches!]

=======Figures=======

5. The colourbars on figures 1, 10 and 11 are extremely difficult to read, with most of the range condensed into a small region on the left and the rest used solely to indicate the extrema in the data. They need to be modified significantly to be useful; saturation in some regions should be an acceptable tradeoff for clarity over most of the globe.

6. Related, most graphs makes heavy use of both red and green; this is difficult for our colourblind colleagues, and should be modified if possible by e.g. changing line styles as well as colours.

7. Figure 7 has the upper panel is labelled in km, and the lower panels in hPa. While the conversions are given in the text, this still makes it hard to read. I'd suggest either

adding a pressure axis to the upper panel or changing the titles of the lower two panels.

8. I'd also suggest putting a box on the maps on figure 7 showing the region covered by figure 6.

9. The black circles on figures 4 and 6 are quite hard to see on my screen; I'd suggest either strengthening or enlarging them.

10. I'd suggest rearranging figures 7 and 8 to not be between figures 6 and 9, as I had to scroll a lot to match up the common features in figures 6 and 9.

11. You refer to both the predicted and directly-estimated precision for both HIRDLS and AIRS for figure 2, but only show one for each. Is there a reason?

12. Use of 'boreal winter 20XX' in several places is ambiguous - is this: December 20XX - February (20XX+1), or December (20XX-1) - February 20XX? It would be clearer to specify it as, e.g. DJF XX/(XX+1), to remove the potential ambiguity.

=======Minor Comments=======

13. I don't understand P05L13 - please rephrase.

14. HIRDLS version 6 is now fairly old, and was supplanted several years ago. Is there a particular reason this was used?

15. P09L30: what height is the 8.1um channel, approximately?

Fritts and Alexander (Rev. Geophys, 2003), doi:10.1029/2001RG000106

Wright et al (Atmos. Chem. Phys., 2017), doi:10.5194/acp-17-8553-2017

Wright et al (Geophys. Res. Lett, 2016), doi:10.1002/2015GL067233
* * *

---

## Author Comment (AC1) · 27 Oct 2017

The comment was uploaded in the form of a supplement:
https://www.atmos-meas-tech-discuss.net/amt-2017-235/amt-2017-235-AC1-supplement.pdf

———————————————————

---

## Author Response (AR1)

**Reply to review comments**

*We thank the reviewers for the time and efforts spent on the manuscript. We considered all comments and hope that the revised draft properly addresses the remaining issues. Please find our point-by-point replies below (colored in blue and in italics).*

**Reviewer #1 Marvin Geller**

This is an excellent paper. The authors have used both AIRS and HIRDLS observations to study atmospheric gravity waves. AIRS is a nadir-viewing instrument, and hence has relatively poor altitude resolution, but AIRS, having cross-track scanning capability, has excellent horizontal resolution. HIRDLS is a limb-viewing instrument that has better vertical resolution, but due to a malfunction has fixed azimuth viewing. Although the two spacecraft on which these instruments are flying, Aqua in the case of AIRS and Aura in the case of HIRDLS have overpasses separated by only a few minutes. The time separation between observations at the same point is actually about 100 minutes, a time separation over which gravity waves can vary considerably, so cases investigated in this paper have been chosen to hopefully minimize the influence of this.

Another point made in these papers is that the high-resolution AIRS retrievals are superior to the operational retrievals for measuring gravity wave variances. The operational retrieval uses 3×3 observational points. This is done to improve retrievals in the presence of clouds, but this is mainly important for the troposphere. The authors show that their high-resolution AIRS retrievals, which use each individual viewing point give superior stratospheric gravity wave information relative to the operational retrievals.

The measure of gravity wave activity used in this paper is gravity wave variance, but to obtain this, the variances due to larger scale atmospheric variability plus the variance due to instrumental noise must be subtracted from the measured variance. This is discussed in considerable detail in the early portions of the paper. Now, the lower altitude resolution and higher horizontal resolution of AIRS relative to HIRDLS means that higher frequency gravity waves will preferentially be seen by AIRS relative to HIRDLS. A point made both early and later in the paper is that these higher frequency waves, with shorter horizontal and longer vertical wavelengths, will carry more momentum than the lower frequency waves seen by HIRDLS, even if the vari- ances seen by the two are similar. The gravity wave variances seen by AIRS and HIRDLS are compared for two cases. The first is for a mountain wave event, and the second is a storm event with active moist convection. For both cases, it is illustrated that the high-resolution AIRS product is superior for sensing gravity wave variances relative to its operational counterpart, and also that the general distribution of gravity wave variances, in both the horizontal and vertical, from the high-resolution

AIRS data closely resembles those of HRDLS, when one takes into account the different frequencies and wavelength sensitivities of AIRS and HRDLS. This certainly suggests the broad-spectrum source nature for gravity waves for both events. Gravity wave variances at 2.5 hPa (about 42 km) show a large correlation with zonal winds at that level for both AIRS and HRDLS. It is interesting that evidence of a similar correlation between winds at 200 hPa and lower stratospheric gravity wave activity was noted by Wang and Geller (2003).

*We thank you for the supporting comments. Citation was added.*

One important conclusion of this paper is that given the superior altitude and vertical scanning capability of HRDLS, which allows estimates of gravity wave momentum fluxes to be made, along with the superior horizontal information from AIRS that results from its horizontal scanning capability, use of the two data sets in a complementary manner should allow gravity wave propagation direction to be inferred by AIRS, and using this information would allow for more certain gravity wave momentum flux information to be derived from HRDLS. Of course, this relies on the broad-spectrum nature of the gravity wave fields emanating from significant gravity wave sources. Since short horizontal and long vertical wavelength gravity waves carry large momentum fluxes, perhaps clever combination of the two data sets can also be used to place more certain bounds on gravity wave momentum fluxes from various sources.

This is a very well written paper, with one exception, and that is the somewhat awkward use of English in a few instances.

*We revised the paper to fix language issues.*

Of course, this is understandable given that only one of the authors is a native English speaker. One example of this is on line 12 on page 1, where the verbal use is "are conform." The term "are similar" would be preferable in my mind.

*Fixed. Thank you.*

This terminology is seen again on line 22 on the same page.

*Fixed. Thank you.*

A similarly awkward terminology is on line 18 of page 12, where the wording "are diverse" is used instead of the more preferable (to me) "are different."

*Fixed. Thank you.*

I also have a couple of relatively minor points that I would like to see dealt with in this paper.

One is a greater emphasis on the implication of broad-spectrum sources of atmospheric gravity waves.

*We add the following paragraph to the introduction:*

*Gravity wave source processes can emit a broad spectrum of waves. For example, it is known that deep convection excites a broad spectrum of gravity wave phase speeds (e.g., Beres et al., 2004), as well as a broad range of gravity wave vertical and, in particular, horizontal wavelengths. There are indications that the horizontal scales range from several ten to several hundred kilometers (e.g., Choi et al., 2012; Trinh et al., 2016; Kalisch et al., 2016; Ern et al., 2017). Similarly, gravity waves emitted from jets and fronts cover horizontal wavelengths from less than 100 km to more than 500 km (e.g., Plougonven and Zhang, 2014, and references therein), and also the horizontal scales of mountain waves cover a range of less than 10 km to several hundred kilometers (e.g., Fritts et al., 2016; Smith et al., 2016; Ehard et al., 2017, and references therein).*

Another is on lines 15 and 16 of page 2, where they point out that satellite observations are only sensitive to a certain portion of the gravity wave spectrum. Of course, this is true for all observational techniques, a point made in Alexander et al. (2010).

*We add the following paragraph:*
*Given the sensitivity limitations of different atmospheric sounding techniques from satellite, it is evident that a single technique is not capable of covering the whole spectral range of atmospheric gravity waves. As has been discussed by, for example, Preusse et al. (2008), or Alexander et al. (2010), a combination of different measurement techniques (for example, a combination of limb, sub-limb, and nadir sounding observations) can help to obtain a more complete picture of the whole spectrum of gravity waves. Still, the range of very short horizontal wavelengths ($< 30$ km) and vertical wavelengths around 5–10 km is not covered by these standard satellite measurement techniques and requires other techniques such as radiosondes or airborne observations (e.g., Fritts et al., 2016).*

I also think the authors might spend a little time pointing out the different vertical phase tilts in the high-resolution AIRS and HRDLS variances in figure 5. This is likely due to the different propagation characteristics of the portion of the gravity wave spectrum seen by the two instruments.

*Hoffmann and Alexander (2009) attributed remaining small differences in the vertical phase structures of the observed waves to the different vertical resolution for both instruments. The lower vertical resolution of AIRS also affects the vertical structure.*

**Anonymous Reviewer #2**

Overall comments: The manuscript presents some interesting and new results on how well gravity wave results from HIRDLS and AIRS high-resolution retrievals agree with each other in sta- tistical averages, and in some individual cases. It also presents informative results that extend and confirm previous conjectures on the complementarity of nadir and

limb measurements, without, however, acknowledging some of that previous work sufficiently. The comparison of AIRS and HIRDLS observational filters is very nice, as are the comparisons of the two data sets for orographic and non-orographic waves, and the comparisons of seasonal patterns of variance. Although a minor point of the paper, the comparison of the gravity wave calculations based on AIRS operational and high-resolution data shows why the latter are needed.

However, the description of the instruments and data used is sometimes unclear, and occasionally wrong or misleading. Similarly, the description of the filtering is also occasionally unclear. The advantages of the filtering they have used, and the differences from alternative methods, is not spelled out.

*Across-track background removal applied to AIRS has the advantage that planetary waves will be largely removed. Remnants of planetary waves may be a problem for methods that use slowly evolving planetary waves obtained by global analysis of the observed temperature field.*
*The planetary wave removal applied to HIRDLS utilizes a global analysis of the observed temperature field, as not enough information is available for local detrending, typically. However, different from those methods, the temporal evolution of even short period traveling planetary waves is explicitly accounted for.*
*For each instrument we selected the detrending methods which were found most suitable in earlier work.*

The wording is sometimes poor or awkward.

*We revised the manuscript to fix language issues.*

Specific Comments:

Sec. 2.1 needs to be revised. The beginning is quite stilted. It could be noted that the 3×3 pattern of AIRS footprints fit within the footprint of the microwave instrument, which is used in the cloud-clearing approach. The discussion of the high- resolution data is needed, but should be made clearer.

*We rephrased the text to make it more clear.*

The source of the pressure mentioned on p. 4, l. 23 is not clear.

*The pressure is calculated based on hydrostatic equilibrium and a given pressure at a reference altitude. The reference pressure at 30 km altitude is obtained from the AIRS operational level-2 data.*

Any additional references for the systematic errors and retrieval diagnostics would be useful if they exist.

*It exists only the reference Hoffmann and Alexander (2009). The retrieval approach and error analysis closely follow Rodgers (2000).*

Do ll 35-36 mean that only nighttime data are used in this study? This seems to be the

case, but it is not clearly stated.

*Yes, only nighttime data are used. We changed the sentence to: The data in this study were split in day- and nighttime depending on the solar zenith angle and only the nighttime data were used.*

The range of the high-resolution retrieval is stated to be 10 to 70 km, with 5–6 degrees of freedom- does this mean that the vertical resolution is 10-12 km?

*The vertical resolution varies between 7–15 km with height.*

In the discussion of HIRDLS, it could be noted that HIRDLS was damaged during launch, precluding its planned ability to scan in azimuth, which would have given it 3D capabilities [Gille et al., 2003]. The damage resulted in its single view direction of -47° relative to the orbit plane. This also required extensive corrections to the processing algorithms [Gille et al., 2008, 2011]. Measurements of thermal emission with 1 km vertical resolution are made in 4 channels on the long-wave side of the 15 $\mu$m bands, from which the temperature is retrieved as a function of pressure Khosravi et al. [2009a,b]. The Field of View of the instrument is always 1 km; the resolution of the retrieval varies with altitude.

*We added your suggestions.*

Sec. 2.2: The "background removal" for AIRS is local, within one cross-track scan, 25° . It is noted that this strongly suppresses wave fronts parallel to the cross-track direction which cover large fractions of each scan. Why isn't this an important problem?

*Revisiting this problem and based on some additional sensitivity tests, we think that our previous wording may have been overemphasizing this specific problem of the AIRS local detrending method. We rephrased the statement as:*
*Note that this procedure tends to suppress wave fronts which are parallel to the across-track direction, but only if the wave patterns covers most of the AIRS measurement track. Smaller scale wave patterns of gravity waves with short along-track wavelengths are typically not affected.*

This seems much different from the method described for HIRDLS. Why couldn't this approach have been applied to comparable data from the overlapping 31 day time windows of HIRDLS data? It would be interesting to see how different those results would be from those used by Fetzer and Gille [1996], Alexander et al. [2008] and Wright et al. [2011, 2013], who used departures from 6 or 7 planetary scale waves that varied smoothly in time. Note that the HIRDLS V6 data also have a gridded product (using a Kalman Filter approach described in Gille et al., 2011).

*For both instruments the well known standard procedures for background removal were applied. Applying the method used by HIRDLS to AIRS would be computationally expensive, because there are 3 million temperature profiles to process each day.*

Please clarify the last sentence of the first paragraph on p. 6. It appears that all small-scale perturbations that get through the filtering are assumed to be GW. Is this correct?

Is there evidence for this assumption?

*This is correct. For HIRDLS only the backgraund variances are removed and no additional noise correction is considered.*

Please comment on last sentence of second paragraph: it is surprising that NH variance in winter is > SH variance in winter, given the large zonal winds and the Andes and Antarctic peninsula as such a large source.

*Please note that this sentence refers to the background variance due to the planetary waves rather than the gravity wave variances.*

Sec. 2.3: HIRDLS empirically estimated precisions for V6 appear to be underesti- mated. The values for V7 are much closer to the predicted precisions up to $\sim 0.5$ hPa, above which they are smaller [Gille et al., 2012].

The last sentence of this section is unclear.

*As can be seen from Figure 2 the predicted HIRDLS temperature noise is quite low, and the bias of temperature variances due to noise is also quite low. Comparing the noise estimate of HIRDLS and AIRS, the values of HIRDLS are quite low and therefore noise is not corrected for in our HIRDLS analysis. We shortened the paragraph to avoid a lengthy and perhaps unnecessary discussion of the HIRDLS noise.*

Sec. 2.4: First paragraph- the treatment of wave phases is not clear.

*We rephrased the paragraph as:*
*Each type of current satellite instruments can detect only a certain part of the full vertical and horizontal wave number spectrum of gravity waves, which is determined by its obser- vational filter (Alexander, 1998; Preusse et al., 2008; Alexander et al., 2010; Trinh et al., 2015). For AIRS the sensitivity to vertical and horizontal wavelengths was determined using an approach similar to Hoffmann et al. (2014). In the vertical direction, tempera- ture profiles representing wave perturbations have been convoluted with the averaging kernel functions of the retrieval to take into account the smoothing effects. In the horizontal di- rection, the polynomial fit detrending method has been applied to given across-track wave perturbation cross-sections to take into account the potential filtering of large-scale features. In both cases, the sensitivity to the given wavelengths was determined by calculating the ratio of the variances of the filtered and unfiltered perturbation data. Here we varied the wave phases over all possible values when we calculated the variances.*

These sensitivity functions, and related discussion, are close to those of Wright et al. ACP 15, 8459- 8477, 2015 [2015], which should be referenced and included in the discussion.

*Citation was added.*

Sec. 4, toward end, could note that some combination of limb and nadir observations was done by Wright et al., GRL 43, 894, 2016.

*Citation was added.*

Figures: Figure 8: Suggest second sentence change to . . .presence of high clouds associated with a storm system. . .

*Fixed. Thank you.*

Technical comments: p.1, l. 3: presumably vertical and horizontal resolution

*Fixed. Thank you.*

l. 12, also

*Fixed. Thank you.*

l. 22- better word than conform needed

*Fixed. Thank you.*

l. 18- better word than "fit" needed

*Fixed. Thank you.*

p.2, l 17 give overviews

*Fixed. Thank you.*

l. 18 comparisons

*Fixed. Thank you.*

p. 3, l. 1 Suggest "Zonal average differences tend to be . . ."

*Fixed. Thank you.*

l. 31 scan covers 1780 km

*Fixed. Thank you.*

p. 5, ll1,2: combine the first 2 sentences

*Fixed. Thank you.*

l. 20: measurement typically consists

*Fixed. Thank you.*

p. 6, l. 28: data are

*Fixed. Thank you.*

p. 7, l. 16: perturbations

*Fixed. Thank you.*

l. 21: The sensitivity function of the current generation of limb sounders. . .

*Fixed. Thank you.*

p. 9, l. 31: Does the sentence beginning in this line refer to Figure 6? Text not clear.

*The sentence refers to Figure 8. We changed the sentence to: Low brightness temperatures indicate the presence of high clouds associated with a storm system in the study area, which could also be a potential source for the gravity wave event.*

p. 12, l. 28: . . .current limb measurements.

*Fixed. Thank you.*

**Anonymous Reviewer #3**

The paper compares the gravity wave detection capabilities of the AIRS nadir sounder and the HIRDLS limb sounder. Reviewer 1 has already described the science area in some detail, so I will not repeat this except to say that the area is of significant current interest and the study is eminently suitable for AMT. Reviewer 2 has already addressed several important technical details, and I agree with him/her that these should be addressed. In particular, I strongly agree with his/her comments that the large differences in background removal method are important, and will discuss this further in my comments below. I also agree with both other reviewers that the language needs some work, although it is generally clear throughout and to a certain extent can be handled in copy-editing. Aside from this minor issue, the paper is well-structured and clear, and I suggest only moderate additional revisions beyond those suggested by Reviewer 1 and 2.

=======Major comments=======

1. I feel that the time difference between the two datasets could do with more consideration. This takes two main forms: 1a. in figures 4,5,6 and 9, the waves appear to be in almost exactly the same phase to the eye. For the mountain wave case, this is quite plausible; however, for the non- orographic case I'd like to see more evidence to confirm why this is so. In particular, since the full three-dimensional wavenumber vector can be inferred from the available data, it should be possible to infer the phase and group velocity of the wave (e.g. Fritts and Alexander 2003; Wright et al 2017), and hence confirm if the change between the two measurement times is indeed so small.

*Nevertheless, the vertical cross-sections of the AIRS high-resolution and HIRDLS retrievals show a similar structure, with larger amplitudes in HIRDLS and slightly larger vertical wavelengths in AIRS. The coarser vertical resolution of AIRS is obvious in the vertical cross-section and results in an attenuation of the amplitudes and coarser vertical structures compared to HIRDLS. This effect increases with altitude, which can be attributed to decreasing vertical resolution of the AIRS retrieval with height. The observed phase shift with altitude is expected, because of the time difference between AIRS and HIRDLS measurements of 100 min and the non-orographic source of the gravity waves.*

1b. in the global time series of variance, presumably there is a not-insignificant time- of-day difference between the two datasets. There's not much that can be done about this, but a little more discussion of how it may affect the results would be useful. This is likely to be most significant in the tropics, where convection has a diurnal cycle: while Aura and Aqua cross the equator in formation, the high viewing angle HIRDLS uses presumably means the scan track will cross at quite a different time than AIRS nadir sensor.

*Yes, indeed, there are local time differences between the two datasets. The main effect, however, is not caused by the the sidewards view of HIRDLS. The main difference is that for AIRS only the descending node is considered (only nighttime data), while for HIRDLS both ascending and descending nodes are considered (daytime data and nighttime data are averaged). This may indeed have some effect in the tropics where a diurnal cycle in the gravity wave sources is expected, but should not have much effect in the polar vortex region during wintertime.*

2a. (also discussed by reviewer 2). The background-removal analysis is inconsistent between the two datasets. I'm not sure why this needs to be so: since global data is available for both AIRS and HIRDLS, presumably a common background removal method could be implemented, presumably more similar to the HIRDLS method used in the paper.

*This could make the two data sets more comparable, but applying the HIRDLS method to AIRS is computational very expensive. We used the well established standard methods for background removal of each instrument. At this point a more detailed comparison of detrending method is beyond the scope of the study.*

2b. also, why in particular is a fourth-order polynomial specifically used for the background removal? I realise this is in common with previous studies, but my understanding was that this was to remove solar glint from the AIRS radiances, which is presumably removed in the 3D temperature retrieval. [I am happy to be corrected on this!]

*For radiances, the general purpose of the 4th order polynomial was to remove large-scale features of any kind from the AIRS observations. This could be effects of the so-called limb-brightening (for the outermost AIRS measurement tracks the path through the atmosphere is longer, and incoming radiances are thus increased), as well as large-scale variations due to changes in the background temperature (e.g. temperature gradients at the polar vortex edge). Of course, the effect of limb brightening should be removed by the temperature retrieval, however, large-scale temperature structures could still bias gravity wave analyses. The use of a 4th-order polynomial turned out to be a good compromise of removing large scale structures and at the same time keeping as much gravity wave signal as possible. Tests using a 2nd-order polynomial showed that not all large-scale features have been removed, in particular near the vortex edge.*

3. P10L21 onwards: You refer in passing to a double-peak in HIRDLS GW variance at 44N in winter 2007, with no attribution, but then explain in detail a similar features in the AIRS data as being due to an SSW. I definitely believe the AIRS feature - for example, the AIRS time series look extremely similar to figure 3 of Wright et al (2010) and it may be

useful to say this - but it seems odd to focus in the text on this relatively small feature of the AIRS time series but not on the (to my eye) much larger change in the HIRDLS series in early 2007. Do you have any idea why the HIRDLS double-peak in early 2007 occurs?

*The double peak in January 2007 is due to a strong warming (Rösevall et al., 2007). The enlarged peak in the HIRDLS data is mainly caused by short vertical and long horizontal wavelength waves that are not visible for AIRS. This becomes clear if Fig. 12 is compared to Fig. 13. The HIRDLS data which are filtered with the AIRS sensitivity function show a strongly reduced second peak which is more similar to the AIRS time series.*
*We adapted the text and included the reference Wright et al. (2010).*

4. The idea of combining limb and nadir sounders has been used previously, for example by Wright et al (GRL, 2016) [and references therein]. It would be useful to mention this in your conclusions, where you suggest that combining limb and nadir datasets for better coverage would be useful. [I realise there are important differences in the two approaches!]

*Citation was added.*

======Figures======

5. The colourbars on figures 1, 10 and 11 are extremely difficult to read, with most of the range condensed into a small region on the left and the rest used solely to indicate the extrema in the data. They need to be modified significantly to be useful; saturation in some regions should be an acceptable tradeoff for clarity over most of the globe.

*We adapted the colourbars, in particular for Figure 1, which was most difficult to read.*

6. Related, most graphs makes heavy use of both red and green; this is difficult for our colourblind colleagues, and should be modified if possible by e.g. changing line styles as well as colours.

*We adapted the graphs by changing the colours and adding different linestyles.*

7. Figure 7 has the upper panel is labelled in km, and the lower panels in hPa. While the conversions are given in the text, this still makes it hard to read. I'd suggest either adding a pressure axis to the upper panel or changing the titles of the lower two panels.

*Fixed. Thank you.*

8. I'd also suggest putting a box on the maps on figure 7 showing the region covered by figure 6.

*Fixed. Thank you.*

9. The black circles on figures 4 and 6 are quite hard to see on my screen; I'd suggest either strengthening or enlarging them.

*Fixed. Thank you*

10. I'd suggest rearranging figures 7 and 8 to not be between figures 6 and 9, as I had to

scroll a lot to match up the common features in figures 6 and 9.

*Fixed. Thank you.*

11. You refer to both the predicted and directly-estimated precision for both HIRDLS and AIRS for figure 2, but only show one for each. Is there a reason?

*We focus now only on the predicted precision due to a comment of reviewer # 2.*

12. Use of boreal winter 20XX in several places is ambiguous - is this: Decem- ber 20XX - February (20XX+1), or December (20XX-1) - February 20XX? It would be clearer to specify it as, e.g. DJF XX/(XX+1), to remove the potential ambiguity.

*Fixed. Thank you.*

=======Minor Comments=======

13. I don't understand P05L13 - please rephrase.

*Fixed. Thank you.*

14. HIRDLS version 6 is now fairly old, and was supplanted several years ago. Is there a particular reason this was used?

*Regarding gravity waves in the altitude range considered, there is not much difference between V006 and V007. Further, V006 has the advantage of a couple of days more data in January 2005.*

15. P09L30: what height is the 8.1um channel, approximately?

*The $8.1\,\mu m$ channel covers a spectral window region. It shows surface emissions or cloud top temperatures.*

[revised manuscript text omitted]